# APOE ε2 is associated with increased tau pathology in primary tauopathy

Na Zhao [1], Chia-Chen Liu[1], Alexandra J. Van Ingelgom[1], Cynthia Linares[1], Aishe Kurti[1],
Joshua A. Knight[1], Michael G. Heckman[2], Nancy N. Diehl[2], Mitsuru Shinohara[1], Yuka A. Martens[1],
Olivia N. Attrebi[1], Leonard Petrucelli[1], John D. Fryer[1], Zbigniew K. Wszolek[3], Neill R. Graff-Radford[3],
Richard J. Caselli[4], Monica Y. Sanchez-Contreras[1], Rosa Rademakers[1], Melissa E. Murray [1],
Shunsuke Koga [1], Dennis W. Dickson [1], Owen A. Ross[1] & Guojun Bu[1]

Apolipoprotein E (*APOE*) ε4 allele is the strongest genetic risk factor for late-onset Alzheimer's disease mainly by modulating amyloid-β pathology. *APOE* ε4 is also shown to exacerbate neurodegeneration and neuroinflammation in a tau transgenic mouse model. To further evaluate the association of *APOE* genotype with the presence and severity of tau pathology, we express human tau via an adeno-associated virus gene delivery approach in human *APOE* targeted replacement mice. We find increased hyperphosphorylated tau species, tau aggregates, and behavioral abnormalities in mice expressing *APOE* ε2/ε2. We also show that in humans, the *APOE* ε2 allele is associated with increased tau pathology in the brains of progressive supranuclear palsy (PSP) cases. Finally, we identify an association between the *APOE* ε2/ε2 genotype and risk of tauopathies using two series of pathologically-confirmed cases of PSP and corticobasal degeneration. Our data together suggest *APOE* ε2 status may influence the risk and progression of tauopathy.

[1] Department of Neuroscience, Mayo Clinic, Jacksonville, FL 32224, USA. [2] Division of Biomedical Statistics and Informatics, Mayo Clinic, Jacksonville, FL 32224, USA. [3] Department of Neurology, Mayo Clinic, Jacksonville, FL 32224, USA. [4] Department of Neurology, Mayo Clinic, Phoenix, AZ 85054, USA. These authors contributed equally: Na Zhao, Chia-Chen Liu. Correspondence and requests for materials should be addressed to C.-C.L. (email: liu.chiachen@mayo.edu) or to G.B. (email: bu.guojun@mayo.edu)

Alzheimer's disease (AD) is a neurodegenerative disorder characterized by the presence of extracellular amyloid plaques and intracellular aggregates of hyperphosphorylated tau protein[1] (encoded by microtubule-associated protein tau gene (*MAPT*)[2]). Apolipoprotein E (apoE) is a major cholesterol carrier in the brain[3–5]. The *APOE* ε4 allele is the strongest genetic risk factor for late-onset AD, increasing brain amyloid burden in an allele dose-dependent manner[3–5]. AD also has extensive tau pathology and is considered a "secondary tauopathy," with primary tauopathies referring to disorders in which tau pathology is not accompanied by amyloid. In AD, tau protein aggregates form neurofibrillary tangles (NFTs) and neuropil threads (NTs), as well as dystrophic neurites in senile plaques. Most tau pathology in AD is in neurons, but in primary tauopathies such as progressive supranuclear palsy (PSP), corticobasal degeneration (CBD), and frontotemporal lobar degenerations with tau pathology, tau aggregates are also found in glia[6,7]. In PSP, glial lesions include tufted astrocytes (TAs) and oligodendroglial coiled bodies (CBs)[8].

As a consequence of the extensive amyloid deposition in AD, it is difficult to evaluate the effect of *APOE* alleles on tau pathology in the setting of AD. To exclude the compounding effects of amyloid-β (Aβ), it is critical to investigate the impact of *APOE* alleles in primary tauopathies such as PSP and CBD, which share clinical and pathological mechanisms of tau dysfunction[9], or use pure tau transgenic mouse model system. It is noteworthy that the major genetic risk for many of the primary tauopathies is the common variation in *MAPT* defined by two major haplotypes, with H1 haplotype being more common in primary tauopathies[10,11]. The H1 haplotype has also been implicated in AD[12] and synucleinopathies[13].

A recent study showed that *APOE* ε4 exacerbates neurodegeneration and neuroinflammation in a tau transgenic mouse model[14]. To further examine the association of *APOE* genotype with the presence and severity of tau pathology, we established a gene delivery approach by which adeno-associated virus (AAV) expressing human tau protein bearing P301L mutation (AAV-Tau$^{P301L}$)[15] was bilaterally injected into the cerebral lateral ventricles of neonatal human apoE-targeted replacement (TR) mice with different isoforms. We found that the apoE2-TR mice expressing human tau (Tau$^{P301L}$-apoE2) exhibited significant increases in hyperphosphorylated tau species, thioflavin S-positive tau aggregates, astrocytosis, and behavioral abnormalities. These effects were smaller in apoE3-TR mice and absent in the apoE4-TR mice. We confirmed that the *APOE* ε2 allele is associated with increased tau pathology in the brains of human PSP patients. Finally, we identified a genetic association between *APOE* ε2/ε2 genotype and the risk of PSP and CBD. Together, our results support a pathogenic risk of *APOE* ε2 gene allele with primary tauopathies.

## Results

### Augmented tau pathology and astrogliosis in Tau$^{P301L}$-apoE2 mice.
To generate a tauopathy mouse model in the background of different human apoE isoforms, we bilaterally injected AAV-Tau$^{P301L}$ into the cerebral lateral ventricles of apoE2-TR, apoE3-TR, and apoE4-TR pups at postnatal day 0[15]. The efficiency of AAV vector transduction and human *MAPT* gene expression were evaluated in mice at 6 months of age. No differences were found in the copy numbers of AAV vectors (Supplementary Fig. 1a) or the levels of *MAPT* mRNA (Supplementary Fig. 1b) in the cortex of Tau$^{P301L}$-apoE2-, apoE3-, and apoE4-TR mice. High levels of human tau expression,

reflected by immunoreactivity for V5-tag, were broadly distributed in Tau$^{P301L}$-apoE mice (Supplementary Fig. 2a, b). No differences in total human tau expression were found in cerebral cortex, hippocampus or amygdala among apoE2-, apoE3-, and apoE4-TR mice (Supplementary Fig. 2c–f). Given that NFT contain abnormally hyperphosphorylated tau protein, we first determined the degree of tau phosphorylation in the apoE-TR mice expressing human Tau$^{P301L}$ at 6 months of age. Importantly, Tau$^{P301L}$-apoE2 mice showed an approximately twofold increase in phosphorylation of tau at Ser202 and Thr205 sites (AT8 immunoreactivity) compared with Tau$^{P301L}$-apoE3 or Tau$^{P301L}$-apoE4 mice (Fig. 1a, b; Supplementary Fig. 3a). Consistently, the phosphorylation of tau at Ser199 site was also increased in Tau$^{P301L}$-apoE2 mice compared with Tau$^{P301L}$-apoE3 or Tau$^{P301L}$-apoE4 mice examined by enzyme-linked immunosorbent assay (ELISA) (Supplementary Fig. 4). To further define tau pathology in these mice, we evaluated the formation of tau aggregates monitored by thioflavin S fluorescence, which is a sensitive method to demonstrate inclusions with amyloid-like properties. Consistently, we found that the thioflavin S-positive NFT immunoreactivity in the neocortex of Tau$^{P301L}$-apoE2 mice was twofold higher than Tau$^{P301L}$-apoE3 and Tau$^{P301L}$-apoE4 mice (Fig. 1a–c; Supplementary Fig. 3b). We did not detect significant differences in the numbers of Nissl-positive cells (Supplementary Fig. 5a–c) and NeuN-positive nuclei (Supplementary Fig. 6a, b) between control and Tau$^{P301L}$ mice suggesting there was no difference in neuronal loss in the two models at 6 months of age.

Several lines of evidence indicate that tau pathology is associated with gliosis[15,16], suggesting a toxic interplay between gliosis and tau pathology in disease progression. We histologically examined astrogliosis with glial fibrillary acidic protein (GFAP) immunohistochemistry in the AAV-Tau$^{P301L}$ mouse model (Fig. 1a; Supplementary Fig. 3c). Significantly elevated expression of GFAP-positive astrocytes was noted in the cortex of Tau$^{P301L}$-apoE2 and Tau$^{P301L}$-apoE3 mice compared with their AAV-GFP controls, whereas no increase was found in the Tau$^{P301L}$-apoE4 mice (Fig. 1d). Iba1 immunoreactivity was significantly reduced in Tau$^{P301L}$-apoE4 mice, which might be due to the morphological changes of the microglia, and not different in Tau$^{P301L}$-apoE2 mice or Tau$^{P301L}$-apoE3 mice compared with their own controls, respectively (Supplementary Fig. 7a, c). The mRNA level of *Aif1* gene showed no difference between control and Tau$^{P301L}$ mice in all apoE genotype groups (Supplementary Fig. 7e). There were no significant differences in the staining of CD68, a marker of phagocytic microglia, nor its mRNA levels, between the control and Tau$^{P301L}$ mice in different apoE genotype groups (Supplementary Fig. 7b, d, f). Thus, these results suggest that the microglia activation is not associated with tau pathology in this model.

### Impaired behavior and synaptic integrity in Tau$^{P301L}$-apoE2 mice.
To determine the effects of apoE isoforms on the behavioral performance of AAV-Tau$^{P301L}$ model, we evaluated exploration and anxiety-related behaviors, as well as learning and memory that are often abnormal in various human tauopathies. In the open-field assay (OFA), apoE-TR mice expressing human Tau$^{P301L}$ did not exhibit any hyperactivity as assessed by total distance traveled and time mobile compared with their controls (Supplementary Fig. 8a, b). Tau$^{P301L}$-apoE2 and Tau$^{P301L}$-apoE3 mice displayed a decreased tendency in exploring the center of the field (Fig. 2a), a typical characteristic of increased anxiety. In the elevated plus maze (EPM) task, Tau$^{P301L}$-apoE2 mice spent an increased amount of time in the open arms, whereas Tau$^{P301L}$-apoE3 and Tau$^{P301L}$-apoE4 mice were not compared with their

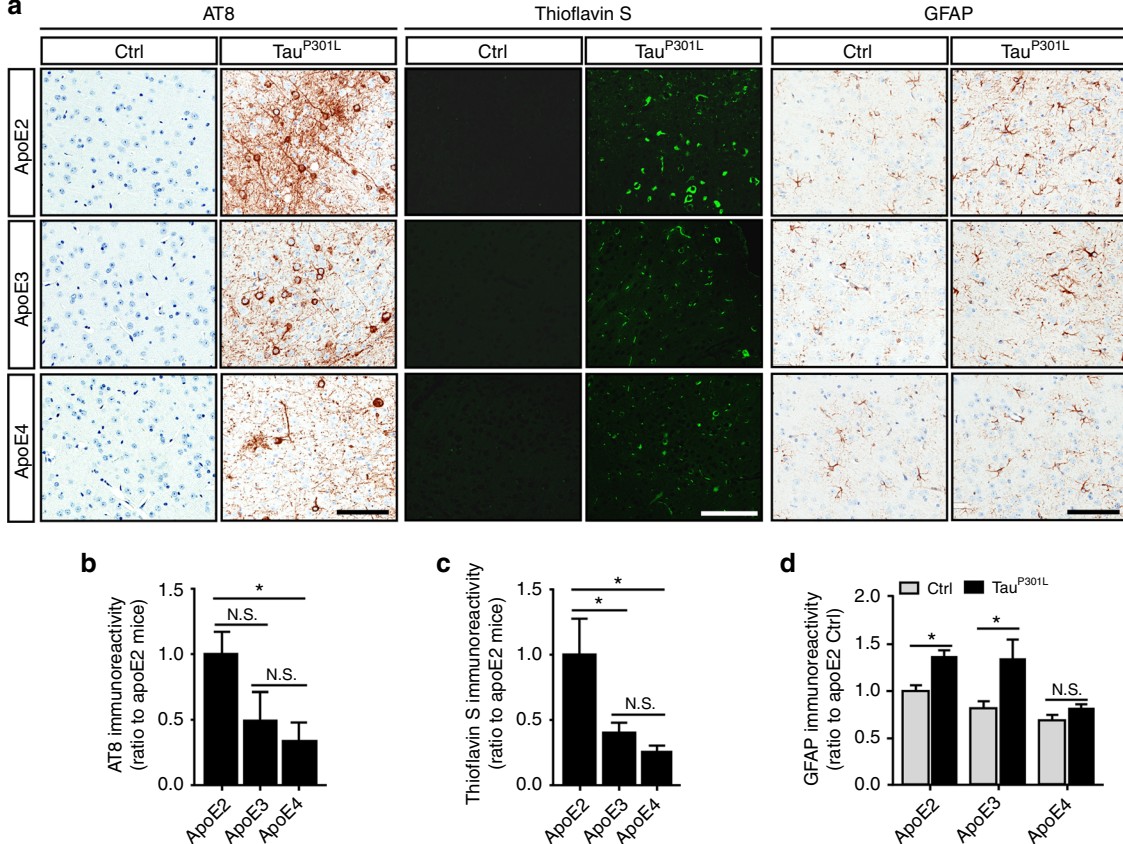

**Fig. 1** Increased hyperphosphorylated tau species, thioflavin S-positive tau aggregates, and related astrogliosis in Tau$^{P301L}$-apoE2 mice. Brain slices from AAV-GFP control mice (Ctrl) and AAV-Tau$^{P301L}$-apoE2, -apoE3, and -apoE4 mice were prepared. The hyperphosphorylated tau, aggregated tau, and astrogliosis were determined by AT8, thioflavin S, and GFAP staining at 6 months of age. **a** Representative images are shown for the deposition of AT8-positive tau species, thioflavin S-positive tau aggregates, and astrogliosis in the cortex (the region above hippocampus) of control (Ctrl) and Tau$^{P301L}$-apoE2, -apoE3, and -apoE4 mice at 6 months of age. Scale bar, 100 μm. The immunoreactivity of AT8 staining (**b**), thioflavin S staining (**c**), and GFAP staining (**d**) was evaluated by Aperio ImageScope ($n = 6$ mice per group, mixed gender). Data represent mean ± SEM. Mann-Whitney tests followed by Bonferroni correction for multiple comparisons were used. *$P < 0.0167$; N.S. not significant

own controls, respectively (Fig. 2b), indicating that Tau$^{P301L}$-apoE2 exhibited aberrant exploratory behavior and disinhibition. Additionally, in a contextual fear conditioning paradigm, Tau$^{P301L}$-apoE2 mice showed significant impairment in the auditory cue-associated memory (Fig. 2d), although there was no significant change in context-associated memory (Fig. 2c). Such impairment was less severe in Tau$^{P301L}$-apoE3 mice and not detectable in Tau$^{P301L}$-apoE4 mice. Taken together, our finding demonstrated that expression of human Tau$^{P301L}$ in the brain of apoE2 mice is detrimental to behavior, to a lesser extent of apoE3 mice, consistent with the observed effects tau pathology in apoE3 and apoE2 mice.

To determine whether synaptic abnormalities were present in Tau$^{P301L}$-apoE mice, we evaluated the expression of the postsynaptic proteins, including GluR2, the subunit of α-amino-3-hydroxy-5-methyl-4-isoxazolepropionic acid receptor (AMPAR), and the postsynaptic density protein 95 (PSD95). The levels of GluR2 (Fig. 2e, f) and PSD95 (Fig. 2e–g) were both significantly decreased in Tau$^{P301L}$-apoE2 mice compared with their AAV-GFP controls, but such effects were not observed in Tau$^{P301L}$-apoE3 and Tau$^{P301L}$-apoE4 mice (Fig. 2e–g).

**Increased insolubility of tau and apoE in Tau$^{P301L}$-apoE2 mice.** We biochemically determined the distribution of soluble and insoluble tau and apoE proteins in the Tau$^{P301L}$-apoE mice by

sequentially extracting cortical brain tissues with RAB (non-detergent-soluble fraction), RIPA (detergent-soluble fraction), and formic acid (FA; detergent-insoluble fraction) buffers. We found that the levels of insoluble tau and apoE were increased in the FA fraction of Tau$^{P301L}$-apoE2 mice (Fig. 3d), while the soluble tau and apoE in both RAB (Fig. 3a, b) and RIPA fractions (Fig. 3a–c) were decreased in Tau$^{P301L}$-apoE2 mice compared with Tau$^{P301L}$-apoE4 mice. The insolubility of tau and apoE protein in Tau$^{P301L}$-apoE3 mice also trended higher compared to that in Tau$^{P301L}$-apoE4 mice although not as pronounced as Tau$^{P301L}$-apoE2 mice. Additionally, the apoE mRNA level was significantly increased in apoE2 mice after Tau$^{P301L}$ expression, whereas no such significant difference was found in apoE3 and apoE4 mice (Supplementary Fig. 9).

It has been reported that tau can bind to apoE3, but not to apoE4, forming a biomolecular complex;[17] however, it remains unclear whether apoE2 can directly form a complex with human tau. To address this, we examined whether recombinant apoE isoforms differentially form a complex with recombinant tau in a cell-free system. Recombinant human apoE protein was incubated with recombinant human tau protein for 1 h at 37 °C in phosphate-buffered saline (PBS). The incubation was ended by addition of sample buffer with or without 2-mercaptoethanol (2-ME), followed by 5 min of boiling before western blotting. In the absence of the reducing agent 2-ME, we observed tau/apoE complexes (bands A and B) that are immune-reactive to both tau

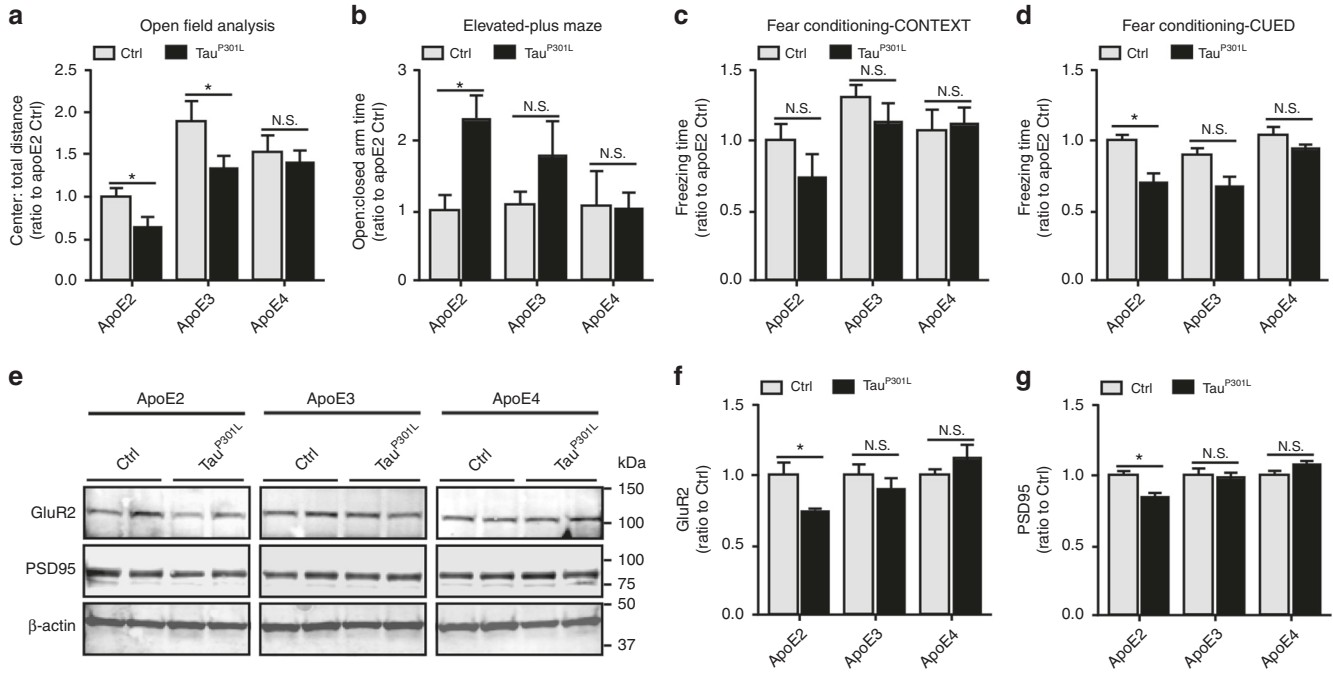

**Fig. 2** Behavioral abnormalities and compromised synaptic integrity in Tau^P301L-apoE2 mice. Behavioral performance was assessed in control mice (Ctrl, AAV-GFP-injected, $n = 12$ mice per group, mixed gender) and Tau^P301L-apoE2, -apoE3, and -apoE4 mice ($n = 13$ mice for Tau^P301L-apoE2 group, $n = 20$ mice for Tau^P301L-apoE3 group, $n = 22$ mice for Tau^P301L-apoE4 group, mixed gender) at 6 months of age. **a** Open-field analysis was assessed and the ratios of time spent in the center quadrant to total distance traveled in the open-field apparatus are shown. **b** Exploratory behavior was evaluated in the elevated plus maze and the ratios of the time spent in open arms to close arms are shown. **c, d** Fear conditioning test was utilized to examine the associative memory. The percentage of the time with freezing behavior in response to stimulus during context (**c**) and cued (**d**) tests is shown. Data represent mean ± SEM. **e–g** The amount of GluR2 and PSD95 in the cortical RIPA lysate was examined by western blotting for control mice and Tau^P301L-apoE mice ($n = 6$ mice per group, mixed gender) at 6 months of age. Results were normalized to β-actin levels. Data are expressed as mean ± SEM. Mann-Whitney tests followed by Bonferroni correction for multiple comparisons were used. *$P < 0.0167$; N.S. not significant

and apoE antibodies (Fig. 4a). Based on stoichiometric calculations, the size of band "A" was consistent with two tau molecules forming a complex with one apoE molecule, while that of band "B" was comprised of one tau and two apoE molecules. ApoE2 formed complexes with tau to a much greater extent than apoE3, whereas such complex was virtually absent when apoE4 was incubated with tau (Fig. 4b, c). The tau/apoE complexes were disrupted in the presence of reducing agent 2-ME (lane 8 and 9), confirming that apoE binds to tau through the formation of intermolecular disulfide bonds. We also incubated the recombinant tau protein with the astrocyte-secreted apoE lipoprotein particles. Similarly, the tau/apoE complexes were only found in the presence of apoE2 or apoE3 lipoprotein particles, but not in apoE4 (Supplementary Fig. 10). However, we did not observe an increase of tau/apoE complex when tau was incubated with apoE2 particles compared with apoE3 particles. We attempted to detect an in vivo interaction between tau and apoE by co-immunoprecipitation studies using brain lysates of our experimental mice. Unfortunately, we could not detect an association between tau and apoE, likely reflecting that such an interaction either does not exist in vivo or is not robust enough for detection by this co-immunoprecipitation method (data not shown). Taken together, our findings suggest that apoE2, non-lipidated form in particular, could form a complex with tau in vitro.

**Increased severity of tau pathology in human PSP patients with *APOE* ε2 allele.** To assess whether *APOE* genotype influences the severity of tau pathology in human PSP patients, we analyzed the association between *APOE* genotype and tau lesion subtypes (CB, NFT, TA, and NT) in PSP brains. The severity of

tau pathology for CB (median score = 1.6), NFT (median score = 2.2), TA (median score = 1.0), and NT (median score = 2.2) is shown in Supplementary Table 1. Consistent with our findings using animal models, after adjustment for multiple testing ($P \leq 0.0125$ considered significant), the presence of *APOE* ε2 was significantly associated with more severe tau pathology (evaluated by overall tau pathology scores) for TA ($P = 0.004$), with similar nominally significant ($P \leq 0.05$) associations observed for CB ($P = 0.045$) and and NT ($P = 0.029$) (Table 1). There was a similar trend for NFT, though it was not quite nominally significant ($P = 0.059$). These findings appear to be driven primarily by the ε2/ε3 genotype, which was more strongly associated with a greater severity of tau pathology for CB ($P = 0.004$), TA ($P = 0.002$), and NT ($P = 0.010$). No significant associations with the severity of tau pathology were noted for the *APOE* ε2/ε2 genotype likely due to an inadequate statistical power for this rare genotype. Given the significant or nominally significant associations between *APOE* ε2 and greater overall tau pathology scores for CB, TA, and NT, we further evaluated the associations of presence of ε2 with the severity of tau pathology for CB, TA, and NT in each separate brain region (Supplementary Tables 2a-c, 3). *APOE* ε2 was most strongly associated with severe CB lesion scores in the inferior olive ($P = 0.003$) and the pontine base ($P = 0.050$), while TA lesions were associated in the basal nucleus of Meynert ($P = 0.001$), globus pallidus ($P = 0.008$), subthalamic nucleus ($P = 0.005$), ventral thalamus ($P = 0.023$), midbrain tectum ($P = 0.001$), and pontine tegmentum ($P = 0.027$), and NT lesions associated with the basal nucleus of Meynert ($P = 0.012$), subthalamic nucleus ($P = 0.018$), medullary tegmentum ($P = 0.023$), and substantia nigra ($P = 0.022$). These results provide further evidence that

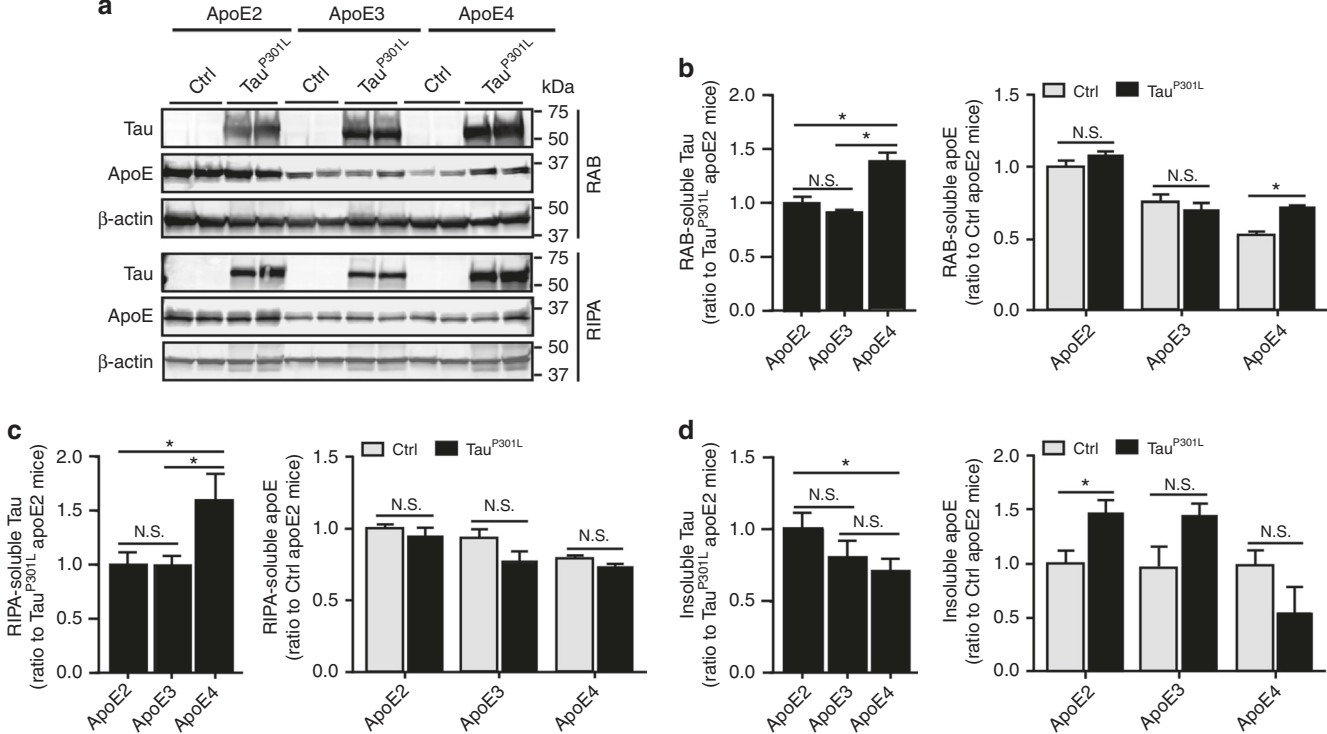

**Fig. 3** Increased insolubility of tau and apoE in Tau[P301L]-apoE2 mice. The cortical brain tissues from control mice (Ctrl, AAV-GFP-injected) and Tau[P301L]-apoE2, -apoE3, and -apoE4 mice at 6 months of age were sequentially extracted by RAB, RIPA, and FA buffer. Soluble tau (HT7 detection) and apoE in RAB (**a**, **b**) and RIPA fractions (**a**, **c**) were examined by western blotting ($n = 6$ mice per group, mixed gender). Results were normalized to $\beta$-actin levels. Insoluble tau and apoE in FA fraction was detected by ELISA (**d** $n = 6$ mice per group, mixed gender). Data are expressed as mean $\pm$ SEM relative to apoE2-TR mice. Mann-Whitney tests followed by Bonferroni correction for multiple comparisons were used. *$P < 0.0167$; N.S. not significant

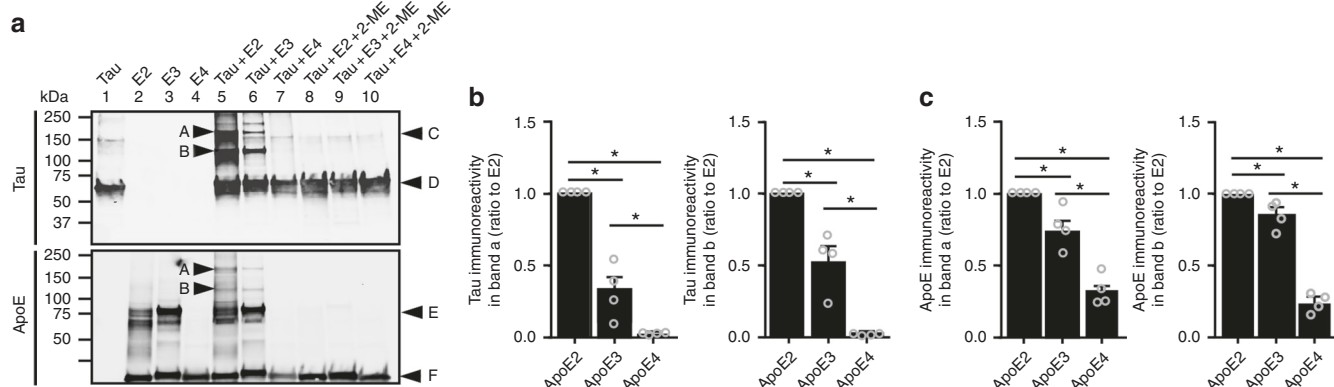

**Fig. 4** Disulfide bond-mediated complex formation between tau and apoE in vitro. The in vitro interaction between tau and apoE proteins was evaluated by solution binding assay, followed by western blotting (four independent experiments). Recombinant human apoE protein was incubated with tau protein for 1 h at 37 °C in 20 µl of phosphate-buffered saline. The reaction was quenched by addition of 20 µl of sample buffer without (lane 1–7) or with (lane 8–10) reducing agent 2-mercaptoethanol (2-ME). **a** The presence of tau/apoE complexes (bands A and B) were determined by immunoblotting with antibodies to tau and apoE, respectively. The amounts of tau (**b**) and apoE (**c**) in the tau/apoE complexes were quantified. Bands A and B were tau/apoE complexes. Band C was a nonspecific band from tau immunoblot. Bands D and F were monomeric tau and apoE, respectively. Band E was estimated to be apoE dimer. Data are expressed as mean $\pm$ SEM relative to apoE2 condition. One-way ANOVA with Tukey post hoc tests were used. *$P < 0.05$; N.S. not significant

*APOE* ε2 affects the severity of tau pathology in human primary tauopathies. As expected, in comparison to the common ε3/ε3 genotype, *APOE* ε4 (in particular the ε4/ε4 genotype) was strongly associated with a higher Thal phase ($P < 0.0001$), while conversely ε2/ε3 was associated with a lower Thal phase (Supplementary Table 4)[18]. No association was found between *APOE* genotype and presence of TDP-43 pathology (Supplementary Table 4).

**Increased risk of PSP and CBD in carriers of the *APOE* ε2/ε2 genotype**. To further investigate the effects of *APOE* genotype on the frequency of tauopathy in humans, we studied 994 PSP patients, 134 CBD patients, and 1406 controls (Supplementary Table 5). As displayed in Table 2, after correction for multiple testing ($P \leq 0.00625$ considered as significant), the only significant association between *APOE* and risk of PSP that we observed occurred for the ε2/ε2 genotype when compared to all other

**Table 1 Associations of *APOE* genotype with tau lesions of PSP**

| APOE genotype (comparisons made vs. the ε3/ε3 reference genotype) | Association with CBs | | Association with NFTs | | Association with TAs | | Association with NTs | |
|---|---|---|---|---|---|---|---|---|
| | Regression coefficient (95% CI) | P-value | Regression coefficient (95% CI) | P-value | Regression coefficient (95% CI) | P-value | Regression coefficient (95% CI) | P-value |
| Presence of ε2 | 0.07 (0.00, 0.13) | 0.045 | 0.06 (0.00, 0.12) | 0.059 | 0.10 (0.03, 0.17) | 0.004 | 0.08 (0.01, 0.16) | 0.029 |
| Presence of ε4 | 0.00 (−0.06, 0.07) | 0.88 | 0.00 (−0.06, 0.06) | 0.99 | 0.04 (−0.03, 0.10) | 0.23 | 0.01 (−0.06, 0.08) | 0.77 |
| Presence of ε2/ε2 | −0.03 (−0.23, 0.16) | 0.74 | 0.10 (−0.08, 0.28) | 0.25 | −0.07 (−0.28, 0.13) | 0.48 | 0.01 (−0.21, 0.23) | 0.92 |
| Presence of ε2/ε3 | 0.11 (0.03, 0.18) | 0.004 | 0.05 (−0.02, 0.12) | 0.15 | 0.12 (0.05, 0.20) | 0.002 | 0.11 (0.03, 0.19) | 0.010 |
| Presence of ε2/ε4 | −0.07 (−0.23, 0.09) | 0.40 | 0.06 (−0.09, 0.21) | 0.41 | 0.11 (−0.06, 0.28) | 0.19 | −0.01 (−0.19, 0.17) | 0.92 |
| Presence of ε3/ε4 | 0.02 (−0.05, 0.08) | 0.62 | 0.01 (−0.05, 0.07) | 0.86 | 0.04 (−0.03, 0.11) | 0.23 | 0.03 (−0.05, 0.10) | 0.47 |
| Presence of ε4/ε4 | 0.03 (−0.17, 0.24) | 0.77 | −0.14 (−0.33, 0.05) | 0.15 | −0.07 (−0.28, 0.14) | 0.51 | −0.11 (−0.34, 0.12) | 0.35 |

Regression coefficients, 95% CIs, and P-values result from linear regression models adjusted for age at death, gender, Braak NFT stage, and Thal amyloid phase. Regression coefficients are interpreted as difference in the mean score of the given neuropathological feature (where scores were calculated by averaging the 0–3 measures across all anatomical structures) between the given *APOE* genotype and the ε3/ε3 reference genotype. P-values ≤ 0.0125 were considered as statistically significant after applying a Bonferroni correction for the four statistical tests that were performed for each different *APOE* genotype categorization
CB coiled body, NFT neurofibrillary tangle, TA tufted astrocyte, NT neuropil thread, CI confidence interval

**Table 2 Associations of *APOE* genotype with risk of PSP and CBD**

| Association (vs. ε3/ε3 unless otherwise noted) | No. (%) in controls | No. (%) in PSP | No. (%) in CBD | Association with PSP (vs. controls) | | Association with CBD (vs. controls) | |
|---|---|---|---|---|---|---|---|
| | | | | OR (95% CI) | P-value | OR (95% CI) | P-value |
| *All subjects* | N = 1406 | N = 994 | N = 134 | | | | |
| Presence of ε2 | 261 (18.6) | 151 (15.2) | 20 (14.9) | 0.75 (0.59, 0.95) | 0.019 | 0.90 (0.53, 1.55) | 0.71 |
| Presence of ε4 | 376 (26.7) | 241 (24.2) | 49 (36.6) | 0.82 (0.67, 1.01) | 0.058 | 1.44 (0.97, 2.16) | 0.071 |
| *APOE genotype* | | | | | | | |
| ε2/ε2 | 5 (0.4) | 13 (1.3) | 2 (1.5) | 4.38 (1.44, 13.33) | 0.009 | 5.24 (0.89, 31.04) | 0.068 |
| ε2/ε3 | 210 (14.9) | 116 (11.7) | 14 (10.5) | 0.69 (0.53, 0.90) | 0.0063 | 0.80 (0.44, 1.47) | 0.47 |
| ε2/ε4 | 46 (3.3) | 22 (2.2) | 4 (3.0) | 0.67 (0.39, 1.18) | 0.17 | 0.94 (0.27, 3.19) | 0.91 |
| ε3/ε3 | 815 (58.0) | 624 (62.8) | 69 (51.5) | 1.00 (reference) | N/A | 1.00 (reference) | N/A |
| ε3/ε4 | 313 (22.3) | 201 (20.2) | 42 (31.3) | 0.81 (0.65, 1.01) | 0.064 | 1.49 (0.98, 2.27) | 0.060 |
| ε4/ε4 | 17 (1.2) | 18 (1.8) | 3 (2.2) | 1.32 (0.65, 2.71) | 0.44 | 1.69 (0.46, 6.24) | 0.43 |
| Presence of ε2/ε2 vs. all other genotypes | 5 (0.4) | 13 (1.3) | 2 (1.5) | 4.41 (1.54, 12.61) | 0.0057 | 4.86 (0.81, 29.01) | 0.083 |
| *Excluding MAPT H1/H1 subjects* | N = 599 | N = 109 | N = 17 | | | | |
| Presence of ε2 | 114 (19.0) | 15 (13.8) | 3 (17.7) | 0.63 (0.34, 1.15) | 0.13 | 1.11 (0.29, 4.22) | 0.88 |
| Presence of ε4 | 157 (26.2) | 26 (23.9) | 5 (29.4) | 0.82 (0.50, 1.35) | 0.44 | 1.32 (0.43, 4.04) | 0.63 |
| *APOE genotype* | | | | | | | |
| ε2/ε2 | 2 (0.3) | 4 (3.7) | 1 (5.9) | 8.09 (1.43, 45.81) | 0.018 | 20.79 (1.56, 277.20) | 0.022 |
| ε2/ε3 | 87 (14.5) | 10 (9.2) | 2 (11.8) | 0.55 (0.27, 1.12) | 0.099 | 0.91 (0.19, 4.31) | 0.90 |
| ε2/ε4 | 25 (4.2) | 1 (0.9) | 0 (0.0) | 0.18 (0.023, 1.37) | 0.098 | N/A | 1.00[a] |
| ε3/ε3 | 353 (58.9) | 69 (63.3) | 9 (52.9) | 1.00 (reference) | N/A | 1.00 (reference) | N/A |
| ε3/ε4 | 125 (20.9) | 24 (22.0) | 5 (29.4) | 0.96 (0.57, 1.60) | 0.87 | 1.62 (0.53, 4.97) | 0.40 |
| ε4/ε4 | 7 (1.2) | 1 (0.9) | 0 (0.0) | 0.67 (0.08, 5.60) | 0.71 | N/A | 1.00[a] |
| Presence of ε2/ε2 vs. all other genotypes | 2 (0.3) | 4 (3.7) | 1 (5.9) | 9.56 (1.71, 53.38) | 0.010 | 21.40 (1.70, 270.53) | 0.018 |

ORs, 95% CIs, and P-values result from logistic regression models adjusted for age, gender, and number of *MAPT* H1 alleles (number of *MAPT* H1 alleles was not adjusted for in analysis excluding *MAPT* H1/H1 subjects). The ε3/ε3 genotype was the reference category for all comparisons unless otherwise noted
OR odds ratio, CI confidence interval
[a]P-value results from Fisher's exact test due to the absence of CBD patients with an ε2/ε4 or ε4/ε4 genotype. P-values ≤ 0.00625 were considered as statistically significant after applying a Bonferroni correction for the eight statistical tests that were performed in the separate PSP vs. controls and CBD vs. controls analyses

genotypes; *APOE* ε2/ε2 was significantly more frequent in PSP compared to controls (1.3% vs. 0.4%, odds ratio (OR) = 4.41, P = 0.0057). Results were similar when comparing ε2/ε2 to a reference ε3/ε3 genotype (OR = 4.38, P = 0.009), although this finding was not quite significant, likely due to the lower power to detect an association with the lower sample size given the similar OR estimate. Interestingly, although not significant, this same trend was also observed in the smaller series of CBD patients, where ε2/ε2 was more common in patients compared to controls (1.5% vs. 0.4%, OR = 4.86, P = 0.083). Noteworthy, although not significant, the ε2/ε3 genotype was associated with a lower risk of PSP (OR [vs. ε3/ε3] = 0.69) suggesting a recessive disease model under the risk paradigm.

The *MAPT* gene has two conserved haplotypes H1 and H2[10], and the common H1 haplotype is significantly over-represented in PSP[10,19,20] and CBD[11,21,22]. To assess the contribution of *APOE* genotype to PSP and CBD susceptibility in the absence of the strong *MAPT* H1 genetic risk factor, we examined the aforementioned associations when excluding individuals who were homozygous for that *MAPT* H1/H1 genotype (Table 2). Notably, the presence of *APOE* ε2/ε2 dramatically increased the risks of both PSP (3.7% vs. 0.3%, OR: 9.56, P = 0.010) and CBD

(5.9% vs. 0.3%, OR: 21.40, P = 0.018) in this subgroup of 109 PSP patients, 17 CBD patients, and 599 controls. Taken together, these data indicate that *APOE* ε2/ε2 genotype increases the risk of both PSP and CBD and may act as a stronger determinant in the absence of *MAPT* H1/H1 status.

## Discussion

In the present study, we demonstrated a significantly increased accumulation of hyperphosphorylated tau species, thioflavin S-positive tau aggregates, and tauopathy-related astrogliosis in a Tau[P301L]-apoE2 disease mouse model, leading to behavioral abnormalities in anxiety, exploration, as well as learning and memory. Furthermore, we showed that *APOE* ε2 was associated with increased severity of tau pathology in the postmortem human brain tissues from PSP patients. Finally, we detected an increased frequency and risk of the recessive *APOE* ε2/ε2 genotype in a large series of PSP and CBD cases.

We introduced mutant Tau[P301L] via viral delivery in humanized mouse models for each *APOE* allele (ε2, ε3, and ε4). In these viral-induced tau mouse models, we found severe tau pathology and behavioral deficits in the presence of *APOE*

ε2/ε2. A recent study showed that *APOE* ε4 exacerbates neurodegeneration and neuroinflammation in a tau transgenic mouse model[14]. In our model system, *APOE* ε4 did not affect tau pathology, behavior, or neuroinflammation. Further, we did not observe neurodegeneration in any of the *APOE* genotypes at 6 months of age, which is consistent with the previously reported phenotypes using AAV-tau as tauopathy mouse model[15]. Thus, it is possible that the discrepancy of *APOE* genotype effects between our model system and that of Shi et al.[14] is due to the different experimental model systems used in each study. For example, the total amounts of tau expressed (AAV-mediated vs. transgenic system), the toxicity of different tau species (P301L vs. P301S mutation), the amount of tau aggregation and pathology within certain time periods (6 vs. 9 months of age), and the absence or presence of tau-mediated neurodegeneration are among the variations between these mouse models. These factors together may contribute to the differences observed between these two model systems. However, our human genetic and pathological data supporting an *APOE* ε2 effect on tauopathy are consistent with the findings from our model studies. Future studies are needed to assess dynamic effects of *APOE* genotype in different model systems, but more importantly to interrogate these findings in human study cohorts.

The apoE protein is primarily expressed by astrocytes in the brain and delivers cholesterol and other lipids to neurons through apoE receptor LRP1 and perhaps LDLR[3]. Several features of apoE2 distinguish it from apoE3 and apoE4 isoforms, which might cause apoE2 to behave differently. ApoE2 has a vastly reduced binding ability to LDLR, contributing to an increased risk of type III hyperlipoproteinemia in *APOE* ε2/ε2 individuals[3]. The level of apoE2 protein is the highest compared with other apoE isoforms in cerebrospinal fluid (CSF)[23], interstitial fluid (ISF)[24], brain parenchyma[25,26], and plasma[26–28]. Regarding amyloid pathology, apoE2 and to a lesser extent apoE3 is hyperlipidated, leading to reduced Aβ aggregation and faster clearance of Aβ from the brain[4]. How apoE isoforms affect tau pathology is unclear. Although tau is a cytoplasmic protein, recent evidence has shown that tau protein is also found in extracellular space, including CSF[29] and brain ISF[30]. Intracellular tau aggregates are in equilibrium with extracellular tau, whose aggregation might mediate spreading of pathologic species of tau between cells[30,31]. Since apoE is naturally a secreted protein and readily present in the extracellular fluids[24,32], it is possible that apoE may form a complex with tau and modulate its metabolism in the brain. Alternatively, apoE has been reported to be present in the cytoplasm, where it can also modulate tau aggregation and related pathologies[33]. The higher amount of apoE2 protein in the brain might also contribute to the increased tau aggregation compared with other apoE isoforms. Additionally, the potentially different complex forming properties between tau and apoE isoforms might impact tau metabolism. The exact mechanisms by which apoE isoforms differentially regulate tau aggregation under normal and pathological conditions require further investigation.

We observed the *APOE* ε2 allele is associated with greater tau burden in PSP brains, indicating that *APOE* ε2 may enhance tau pathology not only in mouse models but also in humans. While it has been reported that *APOE* ε4 is associated with NFT pathology in postmortem human brains whenever Aβ is present, no such association is observed in brains without Aβ[34]. Most patients with PSP have minimal or no Alzheimer-type pathology (median Braak NFT stage of II–III and median Thal amyloid phase of 1), but some PSP patients have concomitant Alzheimer-type pathology[18]. Interestingly, the *APOE* ε4 allele frequency is similar between pure PSP patients with minimal or no Alzheimer-type pathology (11%) and controls, whereas it is significantly

higher in PSP patients with concomitant Alzheimer-type pathology (64%) or with pathologic aging (38%). Also, a recent publication showed that although *APOE* ε4 carriers had more severe amyloid pathology, the total tau burden and the global cognitive impairment index did not differ from *APOE* ε4 non-carriers in 121 autopsy-confirmed PSP patients[35]. These results suggest that *APOE* ε4 is a risk factor for Alzheimer-type amyloid pathology but not necessarily tau burden in PSP[18,36].

*APOE* ε2 is protective in the setting of AD; however, in the absence of amyloid pathology, it has been previously implicated as a risk factor for tau-related neurodegeneration (e.g., NFT-predominant senile dementia[37]). Although AD patients who are *APOE* ε4 carriers have greater tau pathology compared to non-carriers[5], the *APOE* ε4 genotype is not associated with primary age-related tauopathy (PART), which has medial temporal neurofibrillary pathology but no ("definite PART") or only minimal ("possible" PART) Aβ deposition[38], implying that apoE4 might only impact tau pathology in the presence of amyloid. Susceptibility to agyrophilic grain disease, a common aging-related medial temporal primary tauopathy, is variably associated with *APOE* ε2 allele[39,40]. In our current study, we found an increased risk of PSP (and CBD, though not reaching statistical significance) in patients with the recessive *APOE* ε2/ε2 genotype, which given the rarity of this genotype may explain why it was not detected in the previous genome-wide association study (GWAS) of PSP[41]. GWAS has confirmed that the H1 haplotype of the *MAPT* gene is strongly associated with risk for PSP[10,19,20] and CBD[11,21,22]. In our study, after excluding H1/H1 subjects, the presence of *APOE* ε2/ε2 genotype further increased the risk of PSP and CBD, suggesting that the association of *APOE* ε2/ε2 and tauopathy may be independent of *MAPT* H1 haplotype. The combined results of our studies demonstrate that *APOE* ε2/ε2 is related with increased risk and *APOE* ε2 with severity of tau pathology in a relatively common primary tauopathy. We did not observe significant association between *APOE* ε2/ε2 and severity of tau pathology in PSP, which is likely due to the small number of patients with this genotype.

In summary, our study suggests that *APOE* ε2 plays a role in the severity of tau pathology in PSP and CBD, in mouse models and humans. Our results indicate that *APOE* ε2 is not benign with respect to risk for neurodegenerative diseases, which should be taken into consideration in future clinical trial design and early prevention strategies for tau-related disorders.

## Methods

**Animals**. The three *APOE* genotypes ε2, ε3, and ε4 encode three isoforms of human apoE protein, which differ at amino-acid positions 112 and 158[42]. ApoE-TR mice in which murine *Apoe* gene locus is replaced with human *APOE* ε2/ε2, *APOE* ε3/ε3, or *APOE* ε4/ε4 gene were obtained from Taconic. Animals were housed under controlled conditions of temperature and lighting and given free access to food and water. In all, 12–22 mice/genotype were injected with AAV vectors and used for behavioral tests. Six to eight mice/genotype were randomly selected for further biological analysis. Sample sizes were adequately powered to observe the effects on the basis of past experience[43–46]. All animal procedures were approved by the Mayo Clinic Institutional Animal Care and Use Committee and were in accordance with the National Institutes of Health Guide for the Care and Use of Laboratory Animals.

**Intracerebroventricular viral injections in animals**. V5-tagged Tau$^{P301L}$ or green fluorescent protein (GFP) expression plasmids in AAV1 vectors (abbreviated as AAV-Tau$^{P301L}$ and AAV-GFP, respectively) were prepared as described[15]. Briefly, AAV vectors expressing GFP or Tau$^{P301L}$ under the control of the cytomegalovirus enhancer/chicken β-actin promoter were generated by plasmid transfection with AAV helper plasmids in HEK293T cells. The genomic titer of each virus was determined by quantitative PCR. The constructs were sequence-verified using ABI3730 with Big Dye chemistry following manufacturer's protocols (Applied Biosystems, Foster City, CA, USA). ApoE2-, apoE3-, and apoE4-TR pups at postnatal day 0 were intracerebroventricularly injected with 2 μl of AAV-GFP or AAV-Tau$^{P301L}$ viruses into both hemispheres[15,47]. Briefly, newborn pups were cryoanesthetized and subsequently placed on a cold metal plate. A 30-gauge needle

was used to pierce the skull just posterior to bregma and 2 mm lateral to the midline, and AAV was injected into the lateral ventricles. The pups in each breeder were randomly selected to receive either AAV-GFP or AAV-Tau[P301L] viral injections.

**Behavioral tests**. A behavioral battery consisting of OFA, EPM, and contextual and cued fear conditioning (CFC) tests were performed in 6-month-old apoE2-, apoE3-, and apoE4-TR mice injected with AAV-GFP or AAV-Tau[P301L] as described[15]. In OFA test, mice were placed in the center of an open-field arena (40 × 40 × 30 cm, $W \times L \times H$) and allowed to roam freely for 15 min. An overhead camera was used to track movement with AnyMaze software (Stoelting Co., Wood Dale, IL), and mice were analyzed for multiple measures, including total distance traveled, average speed, time mobile, and distance traveled in an imaginary "center" zone (20 × 20 cm). In EPM test, the entire maze is elevated 50 cm from the floor and consists of four arms (50 × 10 cm) with two of the arms enclosed with roofless gray walls (35 × 15 cm, $L \times H$). Mice were tested by placing them in the center of the maze facing an open arm, and their behavior was tracked for 5 min with an overhead camera and AnyMaze software. The CFC test was conducted in a sound attenuating chamber with a grid floor capable of delivering an electric shock, and freezing was measured with an overhead camera and FreezeFrame software (Actimetrics, Wilmette, IL). Mice were initially placed into the chamber undisturbed for 2 min, during which time baseline freezing behavior was recorded. An 80-dB white noise served as the conditioned stimulus (CS) and was presented for 30 s. During the final 2 s of this noise, mice received a mild foot shock (0.5 mA), which served as the unconditioned stimulus (US). After 1 min, another CS–US pair was presented. The mouse was removed 30 s after the second CS–US pair and returned to its home cage. Twenty-four hours later, each mouse was returned to the test chamber and freezing behavior was recorded for 5 min (context test). For the auditory CS test, environmental and contextual cues were changed as described[15]. The animals were placed in the apparatus for 3 min and then the auditory CS was presented and freezing was recorded for another 3 min (cued test). Baseline freezing behavior obtained during training was subtracted from the context or cued tests to control for animal variability.

**Tissue preparation**. Mice were deeply anesthetized with isoflurane prior to transcardially perfused with saline. The brain was removed and bisected along the midline. Half was drop-fixed in 10% neutral buffered formalin (Fisher Scientific, Waltham, MA) overnight at 4 °C for histology, whereas the other half was frozen on dry ice and stored at −80 °C. For biochemical analysis, the brain tissues were homogenized and lysed in RAB buffer [100 mM MES, 1 mM EDTA, 0.5 mM MgSO4, 750 mM NaCl, 20 mM NaF, and 1 mM Na₃VO₄] (G-Biosciences), supplemented with protease inhibitor (Roche) and phosphatase inhibitor (Roche). The samples were centrifuged at 50 000 × $g$ for 20 min at 4 °C. The supernatants were collected as RAB-soluble fractions. The pellets were re-suspended in RIPA (Thermo Fisher Scientific), supplemented by protease inhibitor (cOmplete, Roche) and phosphatase inhibitor (PhosSTOP, Roche) and centrifuged at 50 000 × $g$ for 20 min at 4 °C. The supernatants were collected as RIPA-soluble fractions. The pellets were re-suspended in 70% FA and centrifuged at 50 000 × $g$ for 20 min at 4 °C. The supernatants were collected as FA fractions[48]. All fractions were stored in −80 °C until used for western blot and ELISA analysis.

**Histology and immunohistochemistry**. The half brain fixed in 10% formalin was embedded in paraffin wax, sectioned in a coronal plane at 5 μm thickness and mounted on glass slides. The tissue sections were deparaffinized in xylene and rehydrated in a graded series of alcohols. Antigen retrieval was performed by steaming in distilled water for 30 min, and endogenous peroxidase activity was blocked by incubation in 0.03% hydrogen peroxide. Sections were then immunostained using the DAKO Autostainer (DAKO North America, Carpinteria, CA) and the DAKO EnVision + HRP system. The stained slides were then dehydrated, coverslipped, and scanned with the Aperio Slide Scanner (Aperio, Vista, CA) as described[44]. The following primary antibodies were used: anti-V5 (Cat# R960-25, Thermo Fisher Scientific, 1:500); anti-phospho-tau at site Ser202 and Thr205 (Cat# MN1020, AT8, Thermo Fisher Scientific, 1:250); anti-GFAP (Cat# Pu020-UP, BioGenex, 1:2500); anti-Iba1 (Cat# 019-19741, Wako, 1:2500); anti-CD68 (Cat# ab125212, Abcam, 1:500); and anti-NeuN (Cat# MAB377, clone A60, Millipore, 1:5000) antibodies. For the detection of β-pleated sheets, the sections were incubated with 1% thioflavin S (ThioS, Sigma) for 5 min, washed three times with 70% ethanol and two times with PBS, and then mounted with permanent mounting medium. For the Nissl staining, the paraffin-embedded sections were stained in 0.1% cresyl violet solution for 5 min, then were rinsed quickly in distilled water, and were differentiated in 95% ethyl alcohol for 15 min; finally they were cleared in xylene and mounted with permanent mounting medium. A blinded pathological technician performed histology. Data collection and the quantification of immunoreactivity were performed with the investigators unaware of the sample identities until statistical analyses.

**Western blotting**. Equal amounts of protein from the RAB and RIPA fractions of the homogenized tissue lysates were resolved by SDS-polyacrylamide gel electrophoresis (SDS-PAGE) and transferred to polyvinylidene difluoride (PVDF)

membranes. After the membranes were blocked, proteins were detected with primary antibody. Membrane was probed with horseradish peroxidase (HRP)-conjugated secondary antibody, and detected using the Odyssey infrared imaging system (LI-COR). The following antibodies were used: anti-GluR2 (Cat# MAB397, Millipore, 1:1000); anti-PSD95 (Cat# 3450s, Cell Signaling technology. 1:1000); anti-tau (Cat# MN1000B, HT7, Thermo Fisher Scientific, 1:1000); anti-apoE (Cat# K74180B, Meridian Life Science, 1:1000); and anti-β-actin (Cat# A2228, Sigma, 1:2000) antibodies. The scans of the membrane are shown in Supplementary Fig. 11.

**Enzyme-linked immunosorbent assay**. Levels of apoE and tau in FA fractions and phosphor tau (pS199) in RIPA fraction were determined by ELISAs. For apoE ELISA, WUE4 capture antibody (Cat# NB110-60531, Novus, 1:1000) and biotin-conjugated mouse monoclonal detector antibody (Cat# K74180B, Meridian Life Science, 1:10 000) were used as described[46]. For tau ELISA, a rabbit anti-C terminus of tau capture antibody (Cat# X4000, Osenses, 1:1000) and biotin-conjugated mouse monoclonal HT7 detector antibody were used as described[49]. Recombinant human apoE (Fitzgerald) and human tau proteins (Sigma) were used as standards. The phosphor tau at the site of pS199 was examined by ELISA kit (Cat# KHB7041, Thermo Fisher Scientific) according to the instruction of the manufacturer. Briefly, 50 μl of Standard Diluent Buffer and 50 μl of samples were added into the antibody-coated palte and incubated for 2 h at room temperature. After washing, the plate was incubated with the Detection Antibody solution for 1 h and then HRP solution for 30 min. Colorimetric quantification was performed on a Synergy HT plate reader (BioTek) using HRP-linked streptavidin (Vector) and 3,3′,5,5′-tetramethylbenzidine substrate (Sigma).

**Genomic DNA extraction and AAV copy number analysis**. DNA isolation from cortex of the AAV-TauP301L-apoE-TR mice ($n = 8$ mice per group) was performed using the Puragene kit (Gentra Systems, Minneapolis, MN) in accordance with the manufacturer's protocol. Total DNA concentration was determined by spectrophotometry using Nanodrop, and 10 ng of DNA from each sample was used as the template material for real-time PCR. Real-time PCR was performed on each sample for both the human *MAPT* gene in order to determine copies of the viral genome, and the mouse *Actb* gene to standardize for number of mouse genomes present in each sample. The primers used for human *MAPT* gene were: 5′-TGAA CCAGGATGGCTGAGC-3′ (forward) and 5′-TTGTCATCGCTTCCAGTCC-3′ (reverse). The primers used for *Actb* gene were: 5′-AGTGTGACGTTGACATCC GTA-3′ (forward) and 5′-GCCAGAGCAGTAATCTCCTTC-3′ (reverse). The accumulation of PCR products for each gene was measured using SYBR green (Thermo Fisher Scientific, Waltham, MA). All the samples were run on a CFX96™ Real-Time PCR Detection System (Bio-Rad, Hercules, CA). The absolute amount of each gene was obtained by referring to a standard curve consisting of mouse actin plasmid DNA (Origene, Rockville, MD) or human Tau[P301] plasmid DNA at 10 pg, 5 pg, 1 pg, 100 fg, and 10 fg. All samples and standard plasmid DNA were run in duplicate.

**RNA isolation and real-time PCR analysis**. Total RNA was isolated by using Trizol (QIAGEN, Hilden, Germany), RNeasy Mini Kit (QIAGEN, Hilden, Germany) and subjected to DNase I digestion to remove contaminating genomic DNA. Reverse transcription was performed using iScript™ Reverse Transcription Supermix (Bio-Rad, Hercules, CA). cDNA was added to a reaction mix (20 μl final volume) containing gene-specific primers and SYBR green supermix (Bio-Rad, Hercules, CA). All samples were run in duplicate and were analyzed by using CFX96™ Real-Time PCR Detection System (Bio-Rad, Hercules, CA). Relative gene expression was normalized to actin controls and assessed using the $2^{-\Delta\Delta CT}$ method. Primer sequences are as follows (5′–3′): *Actb*: AGTGTGACGTTGACA TCCGTA (forward) and GCCAGAGCAGTAATCTCCTTC (reverse); human *MAPT*: TGAACCAGGATGGCTGAGC (forward) and TTGTCATCGCTTCCAG TCC (reverse); human *APOE*: TGTCTGAGCAGGTGCAGGAG (forward) and TCCAGTTCCGATTTGTAGG (reverse); *Aif1*: GTCCTTGAAGCGAATGCTGG (forward) and CATTCTCAAGATGGCAGATC (reverse); and *CD68*: ACTTCGG GCCATGTTTCTCT (forward) and GCTGG TAGGTTGATTGTCGT (reverse).

**Purification of lipidated apoE particles from culture medium**. Immortalized mouse astrocytes derived from apoE2-TR, apoE3-TR, and apoE4-TR mice were cultured and conditioned with serum-free medium for 48 h. Conditioned medium was concentrated using Amicon centrifugal filter unit (Millipore), and run through HiTrap Heparin column on an AKTA FPLC system (GE Healthcare). Heparin-bound apoE was eluted with NaCl gradient from 0 to 1 M in Tris buffer. The peak fractions containing apoE that is associated with cholesterol were concentrated as described previously[46]. The concentration of apoE was quantified by apoE ELISA.

**Solution binding of tau and apoE**. Recombinant human Tau-40 protein (1 μg, Sigma) and recombinant apoE protein (1 μg, Fitzgerald) or lipidated apoE particles (1 μg) were incubated for 1 h at 37 °C in 20 μl of PBS (pH 7.4). The incubation was ended by addition of 20 μl of sample buffer with or without 2-ME, followed by 5 min of boiling. Proteins were electrophoretically separated by SDS-PAGE gel and transferred to PVDF membranes. After blocking, the membrane was incubated in

primary antibody overnight. The primary antibodies were monoclonal anti-tau antibody (Cat# MN1000B, HT7, Thermo Fisher Scientific, 1:1000) and monoclonal anti-apoE antibody (Cat# K74180B, Meridian Life Science, 1:1000). Membrane was then probed with HRP-conjugated secondary antibody, and detected using the Odyssey infrared imaging system (LI-COR).

**Statistical analyses for animal study.** In order to ensure that results are valid in the presence of non-normal distributions or differing variances between groups, nonparametric Mann-Whitney tests followed by Bonferroni correction for multiple comparisons were used to compare outcomes between apoE2, apoE3, and apoE4 groups, and also between control and Tau$^{P301L}$ separately for apoE2, apoE3, and apoE4 groups. All statistical tests were two-sided.

**Subjects and statistical methods for tau pathology in PSP.** A total of 858 cases of PSP were obtained from the Mayo Clinic brain bank for neurodegenerative disorders approved by the Mayo Clinic Institutional Review Board. The PSP brains were evaluated with standardized histopathologic methods and phospho-tau immunohistochemistry, and all met neuropathologic criteria for PSP[50]. The 858 PSP cases represent all cases with available semiquantatitive tau pathology measures. Quantitative lesion scores as part of the neuropathologic evaluation (performed by a single observer (D.W.D.)) and are included in this analysis (Supplementary Table 1). Semiquantitative tau pathology measures were assessed on a four-point severity scale (0 = none, 1 = mild, 2 = moderate, and 3 = severe). All sections from all cases were processed in an identical manner with phospho-tau monoclonal antibody (CP13, from Dr. Peter Davies, Feinstein Institute, Long Island, NY) and immunohistochemistry using a DAKO Autostainer. Four lesion types were scored (NFT, CB, TA, and NT) in 17–20 different neuroanatomical regions that are vulnerable to PSP (Supplementary Tables 2a-2c)[51]. An overall score was created for each separate tau pathology measure (NFT, CB, TA, and NT) by calculating the mean of the semiquantitative measures (0, 1, 2, or 3) for each PSP patient across all anatomical regions, where a higher overall score indicates more severe tau pathology. The overall tau pathology scores are displayed in Supplementary Table 1. There was a relatively small amount of missing data for the four tau pathology measures (median: 1.9% missing, range: 0.1–22.0% missing, Supplementary Tables 2a–2c). All cases had assessment of Alzheimer-type pathology with thioflavin S fluorescent microscopy, and based upon the density and distribution of plaques and tangles, Braak NFT stage[52] and Thal amyloid phase[53] were generated for each case as in previous publications[54]. In most of the PSP cases ($N = 795$), TDP-43 pathology was assessed as previously described[55] using a polyclonal antibody to a neoepitope in pathologic TDP-43[56]. Information was collected regarding age (age at death), sex, and APOE genotype (ε2/ε2, ε2/ε3, ε2/ε4, ε3/ε3, ε3/ε4, or ε4/ε4). Genotyping for APOE (rs429358 C/T and rs7412 C/T) was performed using a custom TaqMan Allelic Discrimination Assay on an ABI 7900HT Fast Real-Time PCR system (Applied Biosystems, Foster City, CA, USA). All subjects were non-Hispanic Caucasian and unrelated within and between sample groups.

In the primary analysis, associations of APOE genotype with the four separate overall tau scores were evaluated using multivariable linear regression models that were adjusted for age at death, sex, Braak NFT stage, and Thal amyloid phase. For APOE genotype, this was examined in association analysis as presence of the ε2 allele, presence of the ε4 allele, and also presence of each different APOE genotype, where the ε3/ε3 genotype was the reference group for all comparisons. (i.e. each specific genotype or genotype group was compared against ε3/ε3). Regression coefficients and 95% confidence intervals (CIs) were estimated and are interpreted as the change in the mean tau pathology score corresponding to presence of the given APOE genotype. We used a Bonferroni correction to adjust for the four statistical tests that were performed for each APOE variable, after which P-values of 0.0125 or lower were considered as statistically significant.

In exploratory secondary analysis, for tau pathology measures where there was at least a nominally significant ($P \leq 0.05$) association between the overall score and presence of APOE ε2, we additionally assessed associations between APOE ε2 and the semiquantitative measure of severity of tau pathology from each separate anatomical structure using proportional odds logistic regression models, given the ordinal nature of the semiquantitative tau pathology severity measures. The APOE ε3/ε3 genotype was again used as the reference category. For categories of tau pathology severity measures that had fewer than 10 patients for a given anatomical structure, these were collapsed with more common categories in proportional odds regression analysis. ORs and 95% CIs were estimated and are interpreted as the multiplicative increase in the odds of a more severe tau pathology corresponding to presence of the given APOE genotype. Models were again adjusted for age at death, sex, Braak NFT stage, and Thal amyloid phase. In further secondary analysis, we examined associations of APOE genotype (presence of ε2, presence of ε4, and presence of each different APOE genotype, all vs. the reference ε3/ε3 genotype). with Thal amyloid phase and presence of TDP-43 pathology using proportional odds logistic regression models that were adjusted for age at death, sex, and Braak stage (associations with Thal amyloid phase) and binary logistic regression models that were adjusted for age at death, sex, Braak NFT stage, and Thal amyloid phase (associations with presence of TDP-43 pathology). ORs and 95% CIs were estimated. For associations with Thal amyloid phase, ORs are interpreted as the multiplicative increase in the odds of a higher Thal phase corresponding to the presence of the given APOE genotype. For associations with presence of TDP-43 pathology, ORs are interpreted as the multiplicative increase in the odds of presence of TDP-43 pathology corresponding to presence of the given APOE genotype. All tests were two-sided and statistical analyses were performed using SAS.

**Subjects and statistical analysis for human genetic study.** A total of 994 cases of PSP and 134 cases of CBD were obtained from the Mayo Clinic brain bank for neurodegenerative disorders. They were compared to 1406 neurologically normal clinical control subjects who contributed blood samples with informed consent as part of a research protocol approved by the Mayo Clinic Institutional Review Board. These cases and controls were all those that were available for genetic analysis. The PSP and CBD brains were evaluated with standardized histopathologic methods and phospho-tau immunohistochemistry, and all met neuropathologic criteria for PSP[50] or CBD[57]. Information was collected regarding age (age at death), sex, MAPT genotype (H1/H1, H1/H2, or H2/H2), and APOE genotype (ε2/ε2, ε2/ε3, ε2/ε4, ε3/ε3, ε3/ε4, or ε4/ε4). Genotyping for APOE (rs429358 C/T and rs7412 C/T) and MAPT (rs8070723 A/G) alleles was performed using a custom TaqMan Allelic Discrimination Assay on an ABI 7900HT Fast Real-Time PCR system (Applied Biosystems, Foster City, CA, USA). All subjects were non-Hispanic Caucasian and unrelated within and between sample groups. Characteristics of the PSP and CBD patients, and neurologically normal controls are displayed in Supplementary Table 5.

For primary analysis, after confirming control genotype/allele frequencies were in Hardy-Weinberg equilibrium ($P > 0.05$), the associations of APOE genotype with risk of PSP and CBD (i.e. each disease separately vs. controls) were evaluated using ORs and 95% CIs from logistic regression models that were adjusted for age, sex, and number of MAPT H1 alleles. Initially, we examined associations with risk of PSP and CBD for both presence of the ε2 allele and presence of the ε4 allele. Subsequently, associations with PSP and CBD were examined when considering each APOE genotype separately in order to examine whether there may be specific risk or protective genotypes. The APOE ε3/ε3 genotype was used as the reference category for all comparisons, though we did also examine the association between presence of ε2/ε2 vs. all other genotypes given the strong risk effects for both PSP and CBD that were observed for ε2/ε2. In secondary analysis, all of the aforementioned logistic regression association analysis was also performed in the subset of subjects who did not carry the MAPT H1/H1 genotype (MAPT genotype was not adjusted for in this sub-analysis). In order to adjust for multiple testing in our primary analysis, we applied a Bonferroni correction for the eight statistical tests that were performed in the separate PSP vs. controls and CBD vs. controls analyses, after which P-values of 0.00625 or lower were considered as statistically significant. All statistical tests were two-sided. All statistical analyses were performed using SAS (version 9.2; SAS Institute, Inc., Cary, North Carolina).

## Data availability

All relevant data are available from the authors upon reasonable request.

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

## Acknowledgements

We thank P. Sullivan for developing and contributing the apoE-TR mice, and K. Jansen-West, E. Perkerson, and M. Davis for preparing AAV-Tau[P301L] and AAV-GFP control viruses. We are grateful to L. Rousseau and M. Castanedes-Casey for helping with the preparation of mouse brain slices and immunohistochemical staining. This work was supported by NIH grants R01AG046205, R37AG027924, R01AG057181, R01AG035355, RF1AG051504, and P50AG016574 (to G.B.); R01NS078086 (to O.A.R.); and R35NS097261 (to R.R.); a grant from the Cure Alzheimer's Fund (to G.B.); Mayo Clinic Alzheimer's Disease Research Center (ADRC) (P50AG016574, to D.W.D., G.B. and N.Z.); BrightFocus Fellowship and Alzheimer's Association Research Grant (to C.-C.L.); Mayo Clinic Udall Center of Excellence (NINDS P50NS072187, to Z.K.W., R.R., D.W.D. and O.A.R.); NINDS Tau Center without Walls (U54-NS100693, to L.P., D.W.D., R.R. and O.A.R.); the Mayo Clinic Foundation and the Mayo Clinic Center for Individualized Medicine (to O.A.R.); Mayo Clinic Neuroscience Focused Research Team (to Z.K.W.).

## Author contributions

N.Z., C.-C.L. and G.B. developed the research concept and designed the experiments; N.Z. and C.-C.L. performed the animal experiments; A.V.I. helped with mouse brain lysate aliquot, preparation, and analysis; C.L. contributed to the animal maintenance and brain tissue preparation. A.K. and J.A.K. did the mouse behavioral tests; M.G.H. and N. D. did the statistical analysis for human genetic study; M.S. performed tau ELISA; Y.A.M. purified apoE lipoprotein particles and amplified AAV-Tau[P301L] plasmid; O.N.A. helped with immunostaining quantification; L.P. provided AAV-Tau[P301L] and AAV-GFP; J.D.F. supported the animal behavioral tests; Z.K.W., N.R.G. and R.J.C. collected the patient samples; M.Y.S., R.R. and O.A.R. analyzed the human genetic data; D.W.D., M.E.M. and S.K. provided the human postmortem samples, data, and neuropathologic analyses of

PSP and CBD cases; N.Z., C.-C.L., O.A.R. and G.B. wrote the manuscript with critical inputs and edits by co-authors.

## Additional information

**Competing interests:** The authors declare no competing interests.

