## [Peer Review File · Nature Communications]

Reviewers' comments:

Reviewer #1 (Remarks to the Author):

In this manuscript, the authors present data showing that, in a virus-induced mutant tau expression model, the aggregation of tau, and its associated behavioral deficits, are most impacted by the APOE2 isoform of APOE, as opposed to the more obvious candidate, APOE4. In addition, they present evidence that this phenomenon is replicated in the human tauopathies PSP and CBD. This finding is novel and of high interest to the field, given the importance of APOE biology in neurodegenerative diseases and the surprising finding that APOE2 may be a culprit in tauopathies that are free of Abeta.

The primary weakness of this report is the unique viral mouse model that was used to demonstrate the effects of APOE2 on tau pathology. Although the authors have reported on this model previously (Cook et al., 2015), the model is not complete. Most notably, this model does not display the neurodegenerative effects of the P301L mutation that are displayed in the Tg4510 model, including about a 60 percent decrease in hippocampal CA1 neurons, by about 5.5 months.

Although the authors' observations of greater tau pathology in the virally transduced APOE2 mice is of interest (especially given the supporting human data), this issue of non-toxicity needs to be addressed in more detail. In the recent paper from the Holtzman lab (Shi et al., 2017), which the authors mention in the discussion section, the primary finding was that APOE4 expression was associated with greater APOE4-induced neurodegeneration, likely as a result of APOE4-associated microglial and astrocytic activation in their transgenic APOE/P301S tau mouse. Interestingly, mutant Tau expression does not appear to induce significant microglial activation in any of the APOE mice induced with this model, and astrocytic expression only occurs in the APOE2 and APOE3 mice, a very different finding from that of the Holtzman group. The authors should address why this might be the case in this model. In addition, further data is needed to conclusively show that the APOE4 mice induced with this model are not affected in the same way as was shown by the Holtzman group and to investigate the effect of APOE isoform differences on neurodegeneration in the human patients. Specifically:

- 1) CD68 staining should be performed on sections from each mouse group.
- 2) Nissl staining should be performed in order to confirm the lack of neurodegeneration shown with NeuN staining.
- 3) An analysis of cell loss should be performed on the human tissues in order to compliment the analysis on tau accumulation in Fig. 1 and determine whether the APOE isoforms have differing effects on neurodegeneration in human tauopathy patients.

In addition, for the apoE/tau binding experiments in Figure 3e-g, the source of the apoE proteins are not specified, except to say that they are "recombinant." This likely means that they are not lipidated, but this should be specified either way, as it is very important in order to assess the meaning of the experimental results. If the apoE proteins used were not lipidated, the authors should perform a validation experiment using lipidated versions of apoE2, apoE3 and apoE4, either in its purified form or within astrocyte conditioned media.

Also, the following data should be added to the manuscript in order to increase transparency:

- 1) For Figure 1 whole or hemi brain slice images to display the total AT8, Thioflavin S and GFAP staining in the brain. In addition, the figure legend should specify which part of the cortex the display insets are from.

2) If available, direct comparisons of APOE2, APOE3 and APOE4 levels for all experiments should be included. At present, this data is missing for GFAP, Iba1 and behavioral studies. In addition, please specify in the text whether APOE2 differences are in comparison to APOE3/APOE4 or in comparison to control mice. For example, on page 5, line 23 and page 6, line 13, it should be specified that these APOE2 findings were in comparison to the AAV-GFP control, not in comparison to APOE3 and APOE4.

3) On line 22 of page 9, it should be pointed out that the presence of the E4 allele, as well as APOE3/4 carriers were associated with increased rates of CBD.

4) In table 1, data for each genotype should be included, as done in table 2.

5) Although it is very interesting that beta-mercaptoethanol prevents the binding of tau to apoE2 and apoE3, it is still speculation to say that this is due specifically to the extra cysteine residues in these proteins, as opposed to conformational changes that occur in these proteins, which may expose additional cysteine residues that are necessary for disulfide bonds with tau. Therefore, on page 8, line 12 and on page 12, line 3, the word "likely" to "potentially" or something similar should be included.

Finally, the following corrections in the text are needed:

1) In the first line of the abstract, the statement "largely by modulating amyloid-beta metabolism" is a bit clumsy. Please use a clearer summation of APOE4's effects in AD.

2) On page 3, line 18, please change "pathologic" to "pathological"

3) On page 4, line 4, please change "mouse" to "mice"

4) On page 4, line 4, the term "risky role" is a bit too colloquial.

On the whole, this manuscript offers surprising, but exciting and important findings in regards to APOE2's effect on tau pathology, especially as it relates to primary tauopathies such as PSP and CBD. However, additional data and clarification, most notably in regards to the lack of effect seen by APOE4 on tau-induced neurodegeneration, is required.

Reviewer #2 (Remarks to the Author):

The manuscript by Zhao et al provided evidence from both mouse models and human genetics that ApoE2, traditionally associated with protective effects in AD, is detrimental in primary tauopathies. The strength of the study includes the provocative nature of the finding, the inclusion of human studies, and the high quality of data in general. There are several importance concerns to be addressed:

Major comments:

1) One critical question is whether the elevation of the AT8-positive signal in ApoE2-TR mice is associated with more severe neurodegeneration and functional deficits. The behavioral studies and the analyses of the PSD95 and GluR2 provided some interesting hints that worth more region-specific investigation. The authors showed that at 6-months of age, there is no neuronal loss in any of the AAV-tau-transduced mice regardless of the apoE genotype. In a recently published study by Shi et al, in which apoE4 leads to significantly exacerbated atrophy at 9-months of age without an elevation in AT8-positive inclusion. Thus, it would be really informative for the authors to evaluate neurodegeneration in AAV-tau-transduced mice at older age, e.g. 9 months of age, and focus on the specific regions guided by their behavioral and biochemical analyses.

2) The authors suggest that the increase in AT8 inclusions is related to apoE2's ability to bind to tau more tightly via the disulfide bond. Since ApoE4 lacks the binding ability completely while apoE3 does, one would expect that E4 mice would differ significantly from E3 (and E2) in AT8+ inclusions, which is not the case (Fig. 1a, b, c). This discrepancy needs to be addressed.

Specific comments:

1) It is unclear how authors choose the 16 mice of mixed genotypes to plot the correlation analyses. It would be recommended to indicate the different genotypes with different symbols, and to include all the mice in the other analyses in the correlation analyses. One particular concern is whether the correlation is driven by the genotypes, in which case the relationship between AT8 and GFAP does not provide any new information.

In some cases, the number of animals used in different analyses are different. For example, the behavioral studies have 12–22 mice, but other analyses have 6–8 mice. Clarifications of how the subsets of animals were chosen for each analysis is needed

Reviewer #3 (Remarks to the Author):

The present work proposed by Zhao and colleagues, entitled "APOE epsilon 2 is associated with increased risk for primary tauopathy", is potentially interesting considering that very little is yet known about the potential impact of APOE genotypes on tau neuropathological changes (by contrast with amyloid pathology). In this study, the authors used an approach by intracerebroventricular injection of AAV-tau in APOE2, APOE3 and APOE4 targeted replacement mice in order to investigate the impact of the APOE genetic background on tau pathology and tau-associated cognitive impairment. Additionally, the authors assessed a possible genetic association between APOE genotype with susceptibility to PSP and cortico basal neurodegeneration in human brains. While the approach used to address those questions and the various assays performed are meritorious, the conclusion that "APOE2 increases tau pathology" is an overstatement and important validation experiments need to be added to the study to ascertain the present findings: 1) most of the changes observed in APOE2 mice are also detected in APOE3-TR animals; 2) the authors did not carefully quantify the level of AAV transduction in each experimental group (which is directly correlated with the amounts of tau and tau-associated neurotoxic species), 3) the global amounts of APOE in APOE2-, APOE3- and APOE4-TR mice seem to vary, and therefore it is challenging to determine if the observed effects are related to the nature of each APOE isoform OR to the overall quantity of the protein (the amounts of APOE detected by western blot in Fig. 3 appearing higher in APOE2-TR mice as compared with APOE3 and APOE4). Those concerns definitively weaken the solidity of the data presented, and therefore jeopardize the publication of the study as is, even though the findings have potential to revise our appreciation of the role of APOE towards tau neuropathological changes.

Additional major and minor concerns are reported below.

Major concerns:

1) One of the major issues of the paper is that there is no rigorous evaluation of the AAV-tau transduction levels in APOE2, APOE3 and APOE4-TR mice. Because the efficacy of viral transduction may vary between each experimental group, and because the manifestations of tau pathology evaluated throughout the paper may directly be correlated to the expression levels of tau, it is necessary for the authors to investigate the number of genome copies of vector and the expression level of the recombinant tau gene in each experimental group. The IHC shown in Supp. Fig 1. demonstrating an equivalent staining of total tau in the brains of injected mice is not sufficient to address this point thoroughly. The staining shows saturated neurons filled with tau, but this does not inform the readership about the exact expression level of tau.

2) Throughout the entire manuscript, the authors present data normalized to APOE2-TR, APOE3-TR and APOE4-TR control groups. This normalization is somewhat skewed. It is possible, for example, that APOE4-TR mice show an increased GFAP immunoreactivity at baseline without injection of AAV-tau, and somehow the amount of astrocytic immunoreactivity is already saturated

in those mice without the need for further stressor. If this is the case, the conclusion should not be that "tau-associated neurotoxicity is increased in APOE2 TR mice", but rather that "APOE4-TR are already too compromised to show any additional impact of tau overexpression". It would be best if the authors present their data without normalizing them to the respective "non-injected" controls, therefore allowing the readership to appreciate each parameter of the study at baseline in APOE2, APOE3 and APOE4-TR mice.

3) A few concepts presented in the introduction of the manuscript are somewhat misleading:

- The authors suggest that in primary tauopathies, abnormal inclusions of tau also occur in astrocytes. Are the authors suggesting that the impact of APOE on tau neurotoxic aggregation only is relevant in astrocytes? It is not very clear how those affirmations relate to the findings of the paper, as it appears that the expression of tau driven by AAV is essentially detectable in neurons;
- It is not clear what the authors are implying when they say that "these independent genetic loci may be clinically relevant to susceptibility to tau"... Are they suggesting a linkage between tau and APOE genetic loci, even though those two genes are present on different chromosomes? This entire paragraph needs to be clarified.

4) The authors state that the density of GFAP immunoreactive astrocytes is increased in APOE2-TR after AAV-tau injection as compared with APOE3-TR and APOE4-TR mice. However, the graphs presented in Figure 1d suggest that this is not the case: the difference between the means of GFAP immunoreactivity between the control and injected groups is actually larger for the APOE3-TR as compared with APOE2-TR.

5) The only parameter used to investigate tau pathological changes is the AT8 IHC presented in Fig. 1. This is quite a thin result considering that the quantification of AT8 IHC staining is not really reliable, and a western blot quantification would be more appropriate to show differences between each experimental group. Additionally, the AT8 epitope is detected in physiological conditions and other markers of p-tau need to be included in the analysis (pHF1, ALz50 or MC1 staining for misfolded tau).

6) In general, most of the results observed in APOE2-TR and APOE3-TR are relatively similar (GFAP immunoreactivity, cognitive impairment, etc). The title of the manuscript, which only refers to as the adverse impact of APOE2 on tau pathological changes, is misleading.

7) The decreased level of PSD95 detected in APOE2-TR injected with AAV-Tau is not convincing. Because PSD95 is a structural protein of the post-synaptic compartment, additional validation with other post-synaptic markers should be done.

8) The results presented in Figure 3 are puzzling because it is obvious that the amounts of APOE detected in APOE2-TR mice (control or injected with AAV-tau) are higher than in APOE3 and APOE4-TR mice. This observation raises the concern that the supposed effect of APOE2 on tau-associated neuropathological changes may in fact be the consequence of an increase in the levels of APOE, without any impact of the nature of each variant. While difficult to address, the authors should assess the APOE expression level by qRT-PCR in each mouse line and comment on this potential issue in the discussion.

9) In figure 3e: the use of non-lipidated APOE proteins in the assay may lead to inaccurate results.

10) The major issue for the human data statistical analysis is that the authors did not mention what reference group was used. Did they compare individuals with or without E2, or was the E3/E3 individuals used as references? The latter is more appropriate. Additionally, the CERAD scores should be included as well.

11) Based on the data presented in the Supplemental table 1, it appears that a few individuals actually meet the neuropathological criteria for AD. The authors should either exclude those cases or at least comment on this point.

Minor concerns:

- The description of the AAV vector is very sparse and lousy. In particular, no promoter is mentioned and therefore it is unclear if tau is mostly neuronal or not in the entire study.

- P.3, line 5: "Apolipoprotein E is the principle cholesterol carrier in the brain". Should it be "principal"?

- P.3, line 16: do the authors mean "confounding" instead of "compounding"?

- In Supplementary Fig.3c: there is obviously no correlation between AT8 immunoreactivity and Iba1 immunoreactivity. The graph is unnecessary.
- Figure 3a: The western blot image presented for Tau in the RIPA buffer fraction should be modified, considering that one of the band in the APOE2-TR-AAV-tau group did not properly transfer).
- The statistical analyses for the experimental results are described very superficially. No justification is given, for example, as to why the authors systematically used parametrical tests for comparing the different groups.
- For the human data, the authors should not present the association of tau pathology with E2/E2 genotype (the presence of only one E2 allele suffices, as this concerns more cases). There are too few E2/E2 for the authors to expect any dose effect between one or two E2 alleles.

Reviewer #4 (Remarks to the Author):

This is an extremely important paper that promises to re-calibrate the Alzheimer's APOE field. The conventional wisdom is that E4 is all bad and that E2 is all good. Several labs are working to overexpress E2 with viral transduction with the goal of treating or preventing Alzheimer's. These points represent a deliberate or inadvertent oversimplification of what is really a far more complex situation, as laid out beautifully by Bu and colleagues. Far from being "all good", APOE2 increases the risk for CAA and cerebral hemorrhage in addition to the tauopathy risk described herein by Bu. This was totally glossed over by the recent high impact Nature paper from Holtzman showing an effect of APOE4 on tauopathy that does not require an intermediate step of amyloidosis. The Bu experiments are very well designed and the data are clear and convincing. The interpretations are appropriately cautious and represent the true complexity of the situation. This is the most powerful illustration of the negative effects of APOE2 that I have seen in the 20 yrs since APOE4 was first associated with LOAD by Roses and Pericak-Vance.

Reviewer #1: In this manuscript, the authors present data showing that, in a virus-induced mutant tau expression model, the aggregation of tau, and its associated behavioral deficits, are most impacted by the APOE2 isoform of APOE, as opposed to the more obvious candidate, APOE4. In addition, they present evidence that this phenomenon is replicated in the human tauopathies PSP and CBD. This finding is novel and of high interest to the field, given the importance of APOE biology in neurodegenerative diseases and the surprising finding that APOE2 may be a culprit in tauopathies that are free of Aβ.

---- We thank the Reviewer for recognizing the significance and impact of our work.

The primary weakness of this report is the unique viral mouse model that was used to demonstrate the effects of APOE2 on tau pathology. Although the authors have reported on this model previously (Cook et al., 2015), the model is not complete. Most notably, this model does not display the neurodegenerative effects of the P301L mutation that are displayed in the Tg4510 model, including about a 60 percent decrease in hippocampal CA1 neurons, by about 5.5 months.

---- We agree with the Reviewer that it is important to evaluate potential neuronal loss in our AAV-Tau^{P301L}-apoE mouse models. To address this question, we performed Nissl staining (new Supplementary Fig. 5a-c) and NeuN staining (Supplementary Fig. 6a and b), and found no significant neurodegeneration in our AAV-Tau^{P301L} mouse models at 6 months of age. We also consulted Dr. John Fryer who had published the original work using this AAV-tau mouse models (Cook et al., 2015), and confirmed that there was no neurodegeneration or noticeable neuronal loss observed in their wild-type mice expressing AAV-Tau^{P301L} at 9 months of age (unpublished). It is possible that this viral-mediated tau mouse model does not have neuronal loss until much older ages, which might be advantageous to model progression for age-dependent neurodegenerative diseases. Additionally, it has been reported that the timing and extent of neurodegeneration developed in rTg4510 mice can vary among different cohorts and between laboratories (e.g., some studies reported a ~60% decrease in hippocampal CA1 neurons by about 5.5 months (Ramsden et al., 2005; SantaCruz et al., 2005), whereas another study showed a 43% decrease of neurons in rTg4510 mice between 8 and 12 months (Helboe et al., 2017)). In our current study, we would like to highlight the effects of apoE isoforms on the development of tauopathy in the AAV-Tau^{P301L} models. We currently do not have aged cohort available to thoroughly assess the neurodegeneration status in our AAV-Tau^{P301L}-apoE mice during the timeframe of revision, but are planning to investigate this important question in a future study.

Although the authors' observations of greater tau pathology in the virally transduced APOE2 mice is of interest (especially given the supporting human data), this issue of non-toxicity needs to be addressed in more detail. In the recent paper from the Holtzman lab (Shi et al., 2017), which the authors mention in the discussion section, the primary finding was that APOE4 expression was associated with greater APOE4-induced neurodegeneration, likely as a result of APOE4-associated microglial and astrocytic activation in their transgenic APOE/P301S tau mouse. Interestingly, mutant Tau expression does not appear to induce significant microglial activation in any of the APOE mice induced with this model, and astrocytic expression only occurs in the APOE2 and APOE3 mice, a very different finding from that of the Holtzman group. The authors should address why this might be the case in this model. In addition, further data is needed to conclusively show that the APOE4 mice induced with this model are not affected in the same way as was shown by the Holtzman group and to investigate the effect of APOE isoform differences on neurodegeneration in the human patients.

---- We appreciate the Reviewer for making specific suggestions to improve our manuscript. Please find our point-by-point responses as bellow.

Specifically:

1) CD68 staining should be performed on sections from each mouse group.

---- As suggested, we performed IHC staining for CD68 and examined the expression of CD68 by real-time PCR analysis. We found that there were no significant differences in the levels of CD68-positive microglia between control and Tau^{P301L}-apoE mice regardless of apoE isoforms (new Supplementary Fig. 7b, d and f). We also measured the expression of the *Alf1* gene at the mRNA level and did not detect any significant differences among all groups (new Supplementary Fig. 7e). These results indicate that microglial activation was not associated with the increase in tau pathology observed in our model. In the paper from the Holtzman lab (Shi et al., 2017), it is stated that marked upregulation of a cluster of pro-inflammatory genes was identified by Nanostring analysis; however, only one gene (*HPSE*) was shown to be significantly different between apoE3 and apoE4 tau mice among all genes that were examined (please refer to Shi et al., 2017; Fig. 3b-c). In addition, they found that the most significant difference between apoE3 and apoE4 mice was the activation of CD68-positive microglia (please refer to Shi et al., 2017; Fig. 3e-f). However, we did not detect any differences in the CD68 levels among apoE isoforms in our models.

2) Nissl staining should be performed in order to confirm the lack of neurodegeneration shown with NeuN staining.

---- We have now performed Nissl staining as suggested and did not detect significant differences in the numbers of Nissl-positive cells between any of the groups (Supplementary Fig.5a-c), indicating that expression of human Tau^{P301L} does not lead to neuronal loss in these mice at 6 months of age. As mentioned above, we will further investigate whether the neurodegeneration occurs in AAV-Tau^{P301L}-apoE mice at older ages in a separate study. The discrepancy of *APOE* genotype effects between our AAV-tau model and that of Shi et al. might be due to the different experimental model systems. For example, the amounts of tau that is expressed, the toxicity of different tau species, as well as the amount of tau aggregation and pathology within certain time periods could be different among these mouse models, and they may contribute to the difference in observed neurodegeneration. Thus, it is critical to investigate the effect of apoE isoforms on tau pathology in human tauopathies. Our genetic and neuropathologic data indicate that *APOE* ϵ 2 enhanced tauopathy, which is consistent with the findings in our animal studies. Future studies are needed to assess the effects of *APOE* genotype in different model systems, and more importantly to study these findings in other tauopathies.

3) An analysis of cell loss should be performed on the human tissues in order to complement the analysis on tau accumulation in Fig. 1 and determine whether the APOE isoforms have differing effects on neurodegeneration in human tauopathy patients.

---- We thank the Reviewer for this suggestion. We would like to emphasize that the main objective of the human pathological studies on effects of apoE isoforms on tauopathies in the PSP brains was to further corroborate our findings in mouse models as shown in Fig. 1. For this purpose, we analyzed four types of tau lesions (neurofibrillary tangles, neuropil threads, oligodendroglial coiled bodies, and tufted astrocytes) in 17-20 different neuroanatomical regions

of 858 cases of PSP. We found that *APOE* $\epsilon 2$ allele was associated with increased overall burden of tau pathology in PSP, which is consistent with the findings in our mouse models. All PSP patients in this cohort have neuronal loss in the substantia nigra and subthalamic as a defining characteristic of PSP. PSP cases without such neuronal loss would be considered atypical tauopathies and not included in the analysis. To quantitatively assess neuronal loss in over 800 PSP cases is not a trivial task for us in a limited timeframe and also we feel that it would not materially affect the correlations we found between *APOE* $\epsilon 2$ allele and tau burden from semiquantitative analyses.

In addition, for the apoE/tau binding experiments in Figure 3e-g, the source of the apoE proteins are not specified, except to say that they are “recombinant.” This likely means that they are not lipidated, but this should be specified either way, as it is very important in order to assess the meaning of the experimental results. If the apoE proteins used were not lipidated, the authors should perform a validation experiment using lipidated versions of apoE2, apoE3 and apoE4, either in its purified form or within astrocyte conditioned media.

---- We thank the Reviewer for this suggestion. To further confirm the interaction between apoE and tau, we performed new experiments by incubating the recombinant tau protein with astrocyte-secreted apoE lipoprotein particles (new Supplementary Fig.10). As in the original experiments, tau/apoE complexes were only found in the presence of apoE2 or apoE3 lipoprotein particles, but not apoE4 lipoprotein particles. We did not observe an increase of tau/apoE complexes when tau was incubated with apoE2 compared with apoE3 particles, indicating that the lipidation of apoE2 might alter, to some extent, its ability to interact with tau. We have added these discussion points to the revised text.

Also, the following data should be added to the manuscript in order to increase transparency:

---- We thank the Reviewer for making specific suggestions to improve our manuscript.

1) For Figure 1 whole or hemi brain slice images to display the total AT8, Thioflavin S and GFAP staining in the brain. In addition, the figure legend should specify which part of the cortex the display insets are from.

---- We displayed the hemi brain slice images for the AT8, Thioflavin S, and GFAP staining in our new Supplementary Fig. 3a-c. The parts of cortex displayed in the insets are from the cortical region above hippocampus. We have now specified the region in the Figure legend (Fig. 1) as suggested.

2) If available, direct comparisons of APOE2, APOE3 and APOE4 levels for all experiments should be included. At present, this data is missing for GFAP, Iba1 and behavioral studies. In addition, please specify in the text whether APOE2 differences are in comparison to APOE3/APOE4 or in comparison to control mice. For example, on page 5, line 23 and page 6, line 13, it should be specified that these APOE2 findings were in comparison to the AAV-GFP control, not in comparison to APOE3 and APOE4.

---- We thank the Reviewer for this suggestion. We had put all 6 groups (control and Tau mice from different apoE isoforms) together into one graph for comparison as suggested by Reviewer #3. As mentioned, we observed some differences in baseline levels among control groups in GFAP IHC staining and open field analyses (please see the Figure below). Considering these baseline differences between different apoE genotypes, we believe it is justified to compare

Tau^{P301L}-apoE mice to their own controls (Figures shown in our manuscript). Such a comparison allows us to determine which genotype has the most significant impact on pathology and behavior with Tau^{P301L} expression. For V5, AT8, and Thioflavin S staining, we made direct comparisons between apoE2, apoE3 and apoE4 mice with Tau^{P301L} expression because the control mice did not have any signal. (The baseline is considered the same for all apoE genotypes.) We specify in the text that apoE isoform-dependent findings were in comparison to their own controls, respectively (*Page 5, Line 13; Page 6, Line 3*).

3) On line 22 of page 9, it should be pointed out that the presence of the E4 allele, as well as APOE3/4 carriers were associated with increased rates of CBD.

---- As requested by Reviewer 3 we have revised our analysis such that comparisons of genotypes are made against a $\epsilon 3/\epsilon 3$ reference group (rather than vs. all other genotypes). In this revised analysis, the associations with CBD for $\epsilon 4$ ($P=0.071$) and the $\epsilon 3/\epsilon 4$ genotype ($P=0.060$) are no longer significant (i.e. $P \leq 0.05$). We have now clarified this point in the text (*Page 9, Line 15-17*). It is also worth noting that in our recent CBD GWAS, where we had access to a larger series of autopsy-confirmed CBD, there was no evidence in *APOE* $\epsilon 4$ allele frequency with disease risk.

4) In Table 1, data for each genotype should be included, as done in Table 2.

----We agree with the Reviewer. As suggested, we have examined the associations of each *APOE* genotype (in addition to the presence of $\epsilon 2$ and the presence of $\epsilon 4$) with the four types of PSP tau pathology (CB, NFT, TA, and NT) and included the results in the revised manuscript (Table 1). The comparisons were made against the $\epsilon 3/\epsilon 3$ genotype as suggested by Reviewer #3.

5) Although it is very interesting that beta-mercaptoethanol prevents the binding of tau to apoE2 and apoE3, it is still speculation to say that this is due specifically to the extra cysteine residues in these proteins, as opposed to conformational changes that occur in these proteins, which may expose additional cysteine residues that are necessary for disulfide bonds with tau. Therefore, on page 8, line 12 and on page 12, line 3, the word “likely” to “potentially” or something similar should be included.

---- We thank the Reviewer’s suggestion and have now changed the word from “likely” to “potentially”.

Finally, the following corrections in the text are needed:

---- We thank the Reviewer for making specific suggestions to improve our manuscript.

1) In the first line of the abstract, the statement “largely by modulating amyloid-beta metabolism” is a bit clumsy. Please use a clearer summation of APOE4’s effects in AD.

---- We changed statement to “Apolipoprotein E (*APOE*) ε4 allele is the strongest genetic risk factor for late-onset Alzheimer’s disease mainly by modulating amyloid-β pathology”.

2) On page 3, line 18, please change “pathologic” to “pathological”

---- We had changed it to pathological.

3) On page 4, line 4, please change “mouse” to “mice”

---- We had changed the word “mouse” to “mice” as suggested.

4) On page 4, line 4, the term “risky role” is a bit too colloquial.

---- We thank the Reviewer’s suggestion and had changed this description to “pathogenic risk”.

On the whole, this manuscript offers surprising, but exciting and important findings in regards to APOE2’s effect on tau pathology, especially as it relates to primary tauopathies such as PSP and CBD. However, additional data and clarification, most notably in regards to the lack of effect seen by APOE4 on tau-induced neurodegeneration, is required.

---- We thank the Reviewer for the thoughtful review of our manuscript. We believe our revised manuscript is more clear and improved.

Reviewer #2: The manuscript by Zhao et al provided evidence from both mouse models and human genetics that ApoE2, traditionally associated with protective effects in AD, is detrimental in primary tauopathies. The strength of the study includes the provocative nature of the finding, the inclusion of human studies, and the high quality of data in general. There are several importance concerns to be addressed:

---- We appreciate the Reviewer for recognizing the merit of our work!

Major comments:

1) One critical question is whether the elevation of the AT8-positive signal in ApoE2-TR mice is associated with more severe neurodegeneration and functional deficits. The behavioral studies and the analyses of the PSD95 and GluR2 provided some interesting hints that worth more region-specific investigation. The authors showed that at 6-months of age, there is no neuronal loss in any of the AAV-tau-transduced mice regardless of the apoE genotype. In a recently published study by Shi et al, in which apoe4 leads to significantly exacerbated atrophy at 9-months of age without an elevation in AT8-positive inclusion. Thus, it would be really informative for the authors to evaluate neurodegeneration in AAV-tau-transduced mice at older age, e.g. 9 months of age, and focus on the specific regions guided by their behavioral and biochemical analyses.

---- We thank the Reviewer for this suggestion, which was also raised by Reviewer #1. We did Nissl staining (New Supplementary Fig.5a-c) and NeuN staining (Supplementary Fig. 6a and b) to confirm that our AAV-Tau^{P301L} model mice do not have significant neurodegeneration at 6 months of age. The graph shown in our manuscript is analysis of Nissl and NeuN staining in cortex, but we did not detect neuronal loss in any other brain region, including hippocampus and

amygdala (data not shown). We do agree with the Reviewers that future studies will be needed to assess whether neuronal loss is observed in our AAV-Tau^{P301L}-apoE mice older than 6 months of age. We currently do not have an older cohort for such studies; however, upon consultation with Dr. John Fryer who had published the original work using AAV-Tau^{P301L} mouse model in the mouse apoE background (Cook et al., 2015), and confirmed that no noticeable neuronal loss was found at 9 months of age (unpublished). We discussed in our manuscript that it is possible that the discrepancy of *APOE* genotype effects between our model system and that of Shi et al. are due to the different experimental model systems in each study. In addition, our human genetic and pathological data supporting an *APOE* ϵ 2 effect on tau pathology are consistent with the findings from our models. Future studies are needed to assess dynamic effects of *APOE* genotype in different model systems, but more importantly to interrogate these findings in human cohorts (Page 10, Line 13-22). Interestingly, a recent publication showed that the total tau burden and a global cognitive impairment index did not differ between *APOE* ϵ 4 carriers and noncarriers in 121 cases of autopsy-confirmed PSP patients (Koga et al., 2017). In this paper, the effects of *APOE* ϵ 2 on tau burden and cognitive dysfunction was not examined, but this serves as an additional evidence that *APOE* ϵ 4 genotype may not be associated with worsening of tau pathology in PSP (Page 11, Line 25; Page 12, Line 1-3).

2) The authors suggest that the increase in AT8 inclusions is related to apoE2's ability to bind to tau more tightly via the disulfide bound. Since Apoe4 lacks the binding ability completely while apoE3 does, one would expect that E4 mice would differ significantly from E3 (and E2) in AT8+ inclusions, which is not the case (Fig. 1a, b, c). This discrepancy needs to be addressed. --- We thank the Reviewer for suggestion. We agree with the Reviewer that if apoE3 could bind to tau protein as well as apoE2, AAV-Tau^{P301L}-apoE2 and AAV-Tau^{P301L}-apoE3 mice should have similar levels of tau pathology. However, we did not observe differences in levels of hyperphosphorylated tau species or Thioflavin-S-positive tau aggregates in apoE3 as apoE2 mice. This suggests that differences in interaction between apoE isoforms and tau protein might not be the only mechanism leading to the increased tau pathology in apoE2 mice. For example, higher amounts of apoE2 protein *in vivo* might also contribute to severity of tau pathology in apoE2 mice as suggested by Reviewer #3. However, the exact mechanisms by which apoE isoforms differently regulate tau aggregation under normal and pathological conditions require further investigation. We have discussed this point in our revised manuscript (Page 11, Line 2-3; Page 11, Line 15-16).

Specific comments:

1) It is unclear how authors choose the 16 mice of mixed genotypes to plot the correlation analyses. It would be recommended to indicate the different genotypes with different symbols, and to include all the mice in the other analyses in the correlation analyses. One particular concern is whether the correlation is driven by the genotypes, in which case the relationship between AT8 and GFAP does not provide any new information.

--- We thank the Reviewer for this suggestion. Accordingly, we have now removed the correlation analysis between AT8 and GFAP.

In some cases, the number of animals used in different analyses are different. For example, the

behavioral studies have 12–22 mice, but other analyses have 6–8 mice. Clarifications of how the subsets of animals were chosen for each analysis is needed.

---- We thank the Reviewer for the suggestion. For the biological assessment, we randomly selected 6-8 mice/group from total 12-22 mice/group. We have added this information into the methods section in our revised manuscript (*Page 14, Line 5-7*).

Reviewer #3: The present work proposed by Zhao and colleagues, entitled “APOE epsilon 2 is associated with increased risk for primary tauopathy”, is potentially interesting considering that very little is yet known about the potential impact of APOE genotypes on tau neuropathological changes (by contrast with amyloid pathology). In this study, the authors used an approach by intracerebroventricular injection of AAV-tau in APOE2, APOE3 and APOE4 targeted replacement mice in order to investigate the impact of the APOE genetic background on tau pathology and tau-associated cognitive impairment. Additionally, the authors assessed a possible genetic association between APOE genotype with susceptibility to PSP and cortical basal neurodegeneration in human brains. While the approach used to address those questions and the various assays performed are meritorious, the conclusion that “APOE2 increases tau pathology” is an overstatement and important validation experiments need to be added to the study to ascertain the present findings: 1) most of the changes observed in APOE2 mice are also detected in APOE3-TR animals; 2) the authors did not carefully quantify the level of AAV transduction in each experimental group (which is directly correlated with the amounts of tau and tau-associated neurotoxic species), 3) the global amounts of APOE in APOE2-, APOE3- and APOE4-TR mice seem to vary, and therefore it is challenging to determine if the observed effects are related to the nature of each APOE isoform OR to the overall quantity of the protein (the amounts of APOE detected by western blot in Fig. 3 appearing higher in APOE2-TR mice as compared with APOE3 and APOE4). Those concerns definitively weaken the solidity of the data presented, and therefore jeopardize the publication of the study as is, even though the findings have potential to revise our appreciation of the role of APOE towards tau neuropathological changes.

---- We thank the Reviewer for recognizing the significance and impact of our work. We also appreciate the Reviewer for making specific suggestions to improve our manuscript. Please find our point-by-point responses to Reviewer’s comments/concerns as follows:

Additional major and minor concerns are reported below.

Major concerns:

1) One of the major issues of the paper is that there is no rigorous evaluation of the AAV-tau transduction levels in APOE2, APOE3 and APOE4TR mice. Because the efficacy of viral transduction may vary between each experimental group, and because the manifestations of tau pathology evaluated throughout the paper may directly be correlated to the expression levels of tau, it is necessary for the authors to investigate the number of genome copies of vector and the expression level of the recombinant tau gene in each experimental group. The IHC shown in Supp. Fig 1, demonstrating an equivalent staining of total tau in the brains of injected mice is not sufficient to address this point thoroughly. The staining shows saturated neurons filled with tau, but this does not inform the readership about the exact expression level of tau.

---- We thank the Reviewer for the suggestions. To address this question, we examined the transduction efficiency of AAV vector and the expression level of human *MAPT* in our AAV-Tau^{P301L}-apoE mice at 6 months of age as suggested. No differences were found for either the copy number of AAV vectors (new Supplementary Fig. 1a), or human *MAPT* mRNA levels (new Supplementary Fig. 1b) in the cortex of apoE2-, apoE3-, and apoE4-TR mice. These results, along with the tau IHC staining, indicate that the total human tau expression is similar in apoE2, apoE3 and apoE4 mice.

2) Throughout the entire manuscript, the authors present data normalized to APOE2-TR, APOE3-TR and APOE4-TR control groups. This normalization is somewhat skewed. It is possible, for example, that APOE4-TR mice show an increased GFAP immunoreactivity at baseline without injection of AAV-tau, and somehow the amount of astrocytic immunoreactivity is already saturated in those mice without the need for further stressor. If this is the case, the conclusion should not be that “tau-associated neurotoxicity is increased in APOE2 TR mice”, but rather that “APOE4-TR are already too compromised to show any additional impact of tau overexpression”. It would be best if the authors present their data without normalizing them to the respective “non-injected” controls, therefore allowing the readership to appreciate each parameter of the study at baseline in APOE2, APOE3 and APOE4-TR mice.

---- We thank the Reviewer’s suggestions which also raised by Reviewer #1. We had now organize all the 6 groups (control and Tau mice with different apoE isoforms) together into one graph for comparison without normalizing to their own controls (Please refer to Fig. 1d; Fig. 2a-d, f, g; New Supplementary Fig. 5c; Supplementary Fig. 6b; New Supplementary Fig. 7c-f; Supplementary Fig. 8a and b; and New Supplementary Fig. 9). Although some differences were seen in the baseline (control groups) as we mentioned above to the Reviewer #1 (see above), the findings and conclusions are unchanged from the original.

3) A few concepts presented in the introduction of the manuscript are somewhat misleading:
- The authors suggest that in primary tauopathies, abnormal inclusions of tau also occur in astrocytes. Are the authors suggesting that the impact of APOE on tau neurotoxic aggregation only is relevant in astrocytes? It is not very clear how those affirmations relate to the findings of the paper, as it appears that the expression of tau driven by AAV is essentially detectable in neurons;

---- We thank the Reviewer for the comments. We introduced different types of tau lesions in PSP and CBD, including neuronal lesions, such as neurofibrillary tangles (NFT) and neuropil threads (NT), and glial lesions, such as include tufted astrocytes (TA) and oligodendroglial coiled bodies (CB). The glial lesions are characteristics of primary tauopathies, but not found in AD. Thus, we assessed all four types of tau lesions in ~20 brain regions of 858 PSP cases and found the presence of *APOE* ε2 was significantly associated with more severe tau pathology (evaluated by overall tau pathology scores) for TA (P=0.004), CB (P=0.045), NT (P=0.029) and NFT (P=0.056, trending significant) (Table 1). These results indicate that the impact of *APOE* ε2 on tau pathology is not restricted to astrocytic lesions, but likely a more general effect on both neuronal and non-neuronal cell types. In our models, we only detected abnormal tau in neurons, which is similar to that reported in rTg4510 transgenic mice.

- It is not clear what the authors are implying when they say that “these independent genetic loci may be clinically relevant to susceptibility to tau”... Are they suggesting a linkage between tau

and APOE genetic loci, even though those two genes are present on different chromosomes? This entire paragraph needs to be clarified.

---- We thank the Reviewer for highlighting this possible confusion and have deleted this sentence. We were not suggesting any measure of genetic linkage between these independent loci, rather the potential of an epistatic interaction between the two disease-associated loci. In a disease that has a major genetic risk factor (e.g. *MAPT* H1 in PSP or *APOE* ϵ 4 in AD) other disease-related loci may become more apparent when the analysis is conditioned on the primary risk factor. Thus, it is interesting that we observe the *APOE* ϵ 2 more often with the *MAPT* H2 which removes the major H1 risk effect. However, given the speculative nature of this observation, we have removed this sentence in the revised manuscript.

4) The authors state that the density of GFAP immunoreactive astrocytes is increased in APOE2-TR after AAV-tau injection as compared with APOE3-TR and APOE4-TR mice. However, the graphs presented in Figure 1d suggest that this is not the case: the difference between the means of GFAP immunoreactivity between the control and injected groups is actually larger for the APOE3-TR as compared with APOE2-TR.

---- We thank the Reviewer for the careful reading of our manuscript. We had revised the presentation/analysis of this result by organizing all 6 groups (controls and Tau mice with different apoE isoforms) into one graph as suggested. We also corrected our statement to “Significantly elevated expression of GFAP-positive astrocytes was noted in the cortex of Tau^{P301L}-apoE2 and Tau^{P301L}-apoE3 mice compared with their AAV-GFP controls, whereas no increase was found in the Tau^{P301L}-apoE4 mice (Fig. 1d)” (Page 5, Line 9-11).

5) The only parameter used to investigate tau pathological changes is the AT8 IHC presented in Fig. 1. This is quite a thin result considering that the quantification of AT8 IHC staining is not really reliable, and a western blot quantification would be more appropriate to show differences between each experimental group. Additionally, the AT8 epitope is detected in physiological conditions and other markers of p-tau need to be included in the analysis (PHF1, Alz50 or MC1 staining for misfolded tau).

---- We thank the Reviewer’s suggestions. To further confirm the levels of hyperphosphorylated tau (p-tau) species in our animal models, we performed Western blotting for AT8 (please see figure below). ApoE2 mice had the highest amount of hyperphosphorylated tau species at Ser202 and Thr205 sites as detected by AT8 antibody, which is consistent with result from IHC. We tried various methods to optimize experimental conditions for Western blotting; however, we felt that the AT8 immunoblot was not decisive, and we elected not to include it in the manuscript. Thus, we performed p-tau ELISA to quantify the levels of p-tau in the cortex of AAV-Tau^{P301L}-apoE mice. We found that the p-tau species at Ser199 sites were significantly increased in AAV-Tau^{P301L}-apoE2 mice compared with AAV-Tau^{P301L}-apoE3 and AAV-Tau^{P301L}-apoE4 mice, consistent with our findings by AT8 IHC staining (New Supplementary Figure 4). We agree with the Reviewer that pathological tau can be detected by a number of methods (e.g., conformational epitopes detected by Alz50 and MC1 or phospho-epitopes detected by PHF1). However, AT8 is a widely used method for detecting pathological tau species that has the added advantage over Alz50, MC1 and PHF1 of being commercially available, with lot-to-lot standardization, which is not available for the named antibodies developed by Peter Davies and his colleagues.

6) In general, most of the results observed in APOE2-TR and APOE3-TR are relatively similar (GFAP immunoreactivity, cognitive impairment, etc). The title of the manuscript, which only refers to as the adverse impact of APOE2 on tau pathological changes, is misleading.

---- We thank the Reviewer for the suggestion. In our animal model, we found that apoE2 is associated with significantly increased hyperphosphorylated tau species and tau aggregates. Consistent with this, in our human studies, we found that the *APOE* ε2 allele is associated with increased burden of tau pathology in a large series of autopsy-confirmed PSP. In addition, we identified a significant association between the *APOE* ε2/ε2 genotype and risk of tauopathies using large autopsy-confirmed series of PSP and CBD. In most of the experiments, AAV-Tau^{P301L}-apoE3 did not behave similarly to AAV-Tau^{P301L}-apoE2 mice, but they did have similar phenotypes in activation of GFAP, open field behavioral analysis, and apoE/tau interaction. Interestingly, in our human studies, *APOE* ε2/ε3 genotype was significantly associated with increased tau pathology compared with *APOE* ε3/ε3 genotype, whereas no such association was observed in *APOE* ε2/ε4 or *APOE* ε3/ε4. This suggests that *APOE* ε3 might enhance tau pathology in the presence of *APOE* ε2. Whether there is an interaction between apoE2 and apoE3 on tau pathology needs to be addressed in future studies in mouse models with the *APOE* ε2/ε3 genotype. Taken together, based on the findings from both animal and human studies, we concluded that *APOE* ε2 is associated with an increased risk for primary 4R tauopathy.

7) The decreased level of PSD95 detected in APOE2-TR injected with AAV-Tau is not convincing. Because PSD95 is a structural protein of the post-synaptic compartment, additional validation with other post-synaptic markers should be done.

---- We thank the Reviewer for the comment. To determine whether synaptic abnormalities were present in Tau-apoE mice, we evaluated the expression of the postsynaptic proteins, including GluR2, the subunit of α-amino-3-hydroxy-5-methyl-4-isoxazolepropionic acid receptor (AMPA), and PSD95. The levels of GluR2 (Fig. 2e, f) and PSD95 (Fig. 2e, g) were both significantly decreased in Tau^{P301L}-apoE2 mice compared with their AAV-GFP controls, but such effects were not observed in Tau^{P301L}-apoE3 and Tau^{P301L}-apoE4 mice (Fig. 2e-g).

8) The results presented in Figure 3 are puzzling because it is obvious that the amounts of APOE detected in APOE2-TR mice (control or injected with AAV-tau) are higher than in APOE3 and APOE4-TR mice. This observation raises the concern that the supposed effect of APOE2 on tau-associated neuropathological changes may in fact be the consequence of an increase in the levels of APOE, without any impact of the nature of each variant. While difficult to address, the authors should assess the APOE expression level by qRT-PCR in each mouse line and comment on this potential issue in the discussion.

---- We thank the Reviewer for the comment. Using real-time PCR, we found that *ApoE* mRNA was significantly increased in apoE2 mice upon Tau^{P301L} expression, whereas no such changes

were found in apoE3 and apoE4 mice (Supplementary Fig. 9). This upregulation of *ApoE* expression in apoE2 mice may contribute to increase in tau pathology in AAV-Tau^{P301L}-apoE2 mice. As mentioned, the apoE protein level in the brains of apoE-TR mice is apoE isoform-dependent (apoE2>>E3>E4), which is consistent with previously published results (e.g., Sullivan et al., 2011; Ulrich et al., 2013). The increased amount of apoE2 compared with other apoE isoforms is also found in human plasma and brains (Poirier, 2005). Although it remains unclear why apoE2 is higher compared with apoE3 and apoE4, it is known that apoE2 has a vastly reduced binding ability to LDLR (only 1–2% compared with apoE3), limiting the catabolism of apoE2 protein. We cannot rule out the possibility that higher amounts of soluble apoE2 protein might contribute to tau aggregation compared with other apoE isoforms. We have now added these discussions in the text (*Page 11, Line 2-3; Page 11, Line 15-16*).

9) In figure 3e: the use of non-lipidated APOE proteins in the assay may lead to inaccurate results.

---- We thank the Reviewer for the comment, which was also raised by Reviewer #1. As we described above, we performed experiments to examine interaction of tau and apoE using astrocyte-secreted apoE lipoprotein particles. Tau/apoE complexes were found in the presence of apoE2 or apoE3 lipoprotein particles, but not apoE4 lipoprotein particles (new Supplementary Fig. 10). In addition, we did not observe an increase of tau/apoE complexes when tau was incubated with apoE2 compared with apoE3 particles, indicating that lipidation of apoE2 might, to some extent, impact its ability to interact with tau.

10) The major issue for the human data statistical analysis is that the authors did not mention what reference group was used. Did they compare individuals with or without E2, or was the E3/E3 individuals used as references? The latter is more appropriate. Additionally, the CERAD scores should be included as well.

---- We thank the Reviewer for this comment. We agree with the Reviewer and have revised our analysis/presentation of all results so that all genotype comparisons were made against the $\epsilon 3/\epsilon 3$ reference group. Additionally, we compared the $\epsilon 2/\epsilon 2$ genotype with all other genotypes in the disease-risk analysis given the strong odds ratio estimates observed for this specific genotype. Unfortunately we do not have data regarding CERAD scores; however, we feel that the adjustments for Braak NFT stage and Thal amyloid phase that were made in our analysis of associations of *APOE* genotype with coiled bodies, neurofibrillary tangles, tufted astrocytes, and neuropil threads (Table 1) should sufficiently control for the confounding effect of AD-pathology.

11) Based on the data presented in the Supplemental table 1, it appears that a few individuals actually meet the neuropathological criteria for AD. The authors should either exclude those cases or at least comment on this point.

---- We thank the Reviewer for the suggestion. We did not exclude cases with concomitant Alzheimer type pathology, but we did adjust for Braak NFT stage and Thal amyloid phase, which should sufficiently control for confounding effects of concomitant Alzheimer type pathology.

Minor concerns:

- The description of the AAV vector is very sparse and lousy. In particular, no promoter is mentioned and therefore it is unclear if tau is mostly neuronal or not in the entire study.

---- We mentioned in the method section that “AAV vectors expressing GFP or Tau^{P301L} under the control of the cytomegalovirus enhancer/chicken β -actin promoter”. Cytomegalovirus enhancer/chicken β -actin is a strong synthetic promoter (not cell type specific) frequently used to drive high levels of gene expression in mammalian expression vectors. Importantly, the biodistribution of AAV vector delivered to the neonatal mouse brain is dramatically altered by the timing of injection. The injection on neonatal day P0 results in widespread CNS expression and mostly neuronal transduction, whereas administration in later periods of development (24–84 hours postnatal) give more limited biodistribution and non-neuronal transduction (Chakrabarty et al., 2013). In our study, we limited injections to P0, which leads to widespread tau expression in the whole brain (as shown in Supplementary Fig. 2a) and mostly in neurons (as shown in Supplementary Fig. 2b).

- P.3, line 5: “Apolipoprotein E is the principle cholesterol carrier in the brain”. Should it be “principal”?

---- We thank the Reviewer’s for pointing this point. We have changed the word from “principle” to “major”.

- P.3, line 16: do the authors mean “confounding” instead of “compounding”?

---- We mean confounding here.

- In Supplementary Fig.3c: there is obviously no correlation between AT8 immunoreactivity and Iba1 immunoreactivity. The graph is unnecessary.

---- We thank the Reviewer’s comment and have now removed the graph that shows correlation of AT8 and Iba1 immunoreactivity.

- Figure 3a: The western blot image presented for Tau in the RIPA buffer fraction should be modified, considering that one of the bands in the APOE2-TR-AAV-tau group did not properly transfer).

---- We thank the Reviewer’s suggestion and have now modified this figure.

- The statistical analyses for the experimental results are described very superficially. No justification is given, for example, as to why the authors systematically used parametrical tests for comparing the different groups.

---- We thank the Reviewer for pointing this out and have included more detailed information on statistical methods in our revised manuscript. We have used nonparametric statistical tests to ensure that results are valid in the presence of any departures from normality or differing variances between groups. Specifically, nonparametric Mann-Whitney tests followed by Bonferroni correction for multiple comparisons were used to compare outcomes between apoE2, apoE3 and apoE4 groups, and between control and Tau^{P301L} in different apoE isoform groups. All statistical tests were two-sided. Details of statistical methods are presented in figure legends.

- For the human data, the authors should not present the association of tau pathology with E2/E2 genotype (the presence of only one E2 allele suffices, as this concerns more cases). There are too few E2/E2 for the authors to expect any dose effect between one or two E2 alleles.

---- We agree with the Reviewer that the number of subjects with *APOE* $\epsilon 2/\epsilon 2$ genotype is likely too small to identify any major difference in severity of tau pathology for this genotype. Due to inadequate statistical power for this rare genotype, no significant association was noted for the *APOE* $\epsilon 2/\epsilon 2$ genotype. However, Reviewer #1 requested us to include analysis for each *APOE* genotype in Table 1 (i.e. all *APOE* genotypes were in comparison to $\epsilon 3/\epsilon 3$ reference group) such that the analysis performed is more in line with what was shown in Table 2. We do appreciate the Reviewer for pointing this out; thus, in our revised manuscript we have highlighted in the text that due to the rare frequency of the $\epsilon 2/\epsilon 2$ genotype, the statistical power is inadequate to examine the association of this genotype with the severity of tau pathology (*Page 8, Line 14-16*).

Reviewer #4: This is an extremely important paper that promises to re-calibrate the Alzheimer's APOE field. The conventional wisdom is that E4 is all bad and that E2 is all good. Several labs are working to overexpress E2 with viral transduction with the goal of treating or preventing Alzheimer's. These points represent a deliberate or inadvertent oversimplification of what is really a far more complex situation, as laid out beautifully by Bu and colleagues. Far from being "all good", APOE2 increases the risk for CAA and cerebral hemorrhage in addition to the tauopathy risk described herein by Bu. This was totally glossed over by the recent high impact Nature paper from Holtzman showing an effect of APOE4 on tauopathy that does not require an intermediate step of amyloidosis. The Bu experiments are very well designed and the data are clear and convincing. The interpretations are appropriately cautious and represent the true complexity of the situation. This is the most powerful illustration of the negative effects of APOE2 that I have seen in the 20 yrs since APOE4 was first associated with LOAD by Roses and Pericak-Vance.

---- We greatly appreciate the Reviewer for the overall enthusiasm and satisfaction of our work!

Reviewers' comments:

Reviewer #1 (Remarks to the Author):

In the revised manuscript, the authors have gone to great lengths to address reviewers concerns and have provided detailed and clear explanations for all of the changes that were made to the original manuscript. While these efforts are appreciated and the manuscript is improved, there remain a few points of concern. Most importantly, while the data and conclusions are interesting, the major finding that APOE2 mice display more tau pathology using this AAV-P301L injection model is not as impactful as the title of the paper implies. Most importantly, this model is unique in that the virus-induced P301L does not cause neurodegeneration, which is a major feature of most tauopathy mouse models and of course, of human tauopathies. Therefore, while the ability of APOE2 to increase tau aggregation and phosphorylation is intriguing, these studies may be unintentionally missing a more important contribution to tauopathy played by APOE (and most notably APOE4) in terms of their effects on neurodegeneration, as were highlighted by the recent Shi et al. paper from the Holtzman lab. Furthermore, while there are interesting points of corroboration within the human data, I have concerns that the paper is making a bolder statement about APOE2's contribution to human tauopathies than the data actually reveals. This is certainly interesting work that raises very important questions about the role of APOE2 in tau pathology, but the actual data may not merit the level of attention that this story is likely going to garner if it is published in Nature Communications.

I recommend the following revisions:

- 1) The title should be modified to highlight the role of APOE2 in increasing tau pathology, as opposed to increasing the "risk for primary tauopathy." As is, the title makes it sound like APOE2 has been discovered as a risk factor gene for tauopathy, which is clearly not the case. The authors did find an association between APOE2/2 carriers and PSP, and an increase in tau pathology in APOE2/3 carriers with PSP. However, the fact that APOE2/3 carriers also appear to have a lower risk of developing PSP (despite having greater tau burden than APOE3/3 carriers with PSP) is somewhat contradictory. I also question whether the number of APOE2/2 carriers used in this study is sufficient to draw definitive conclusions about the susceptibility of APOE2/2 carriers to tauopathies.
- 2) In Table 2, the authors should include the data comparing each genotype to all other genotypes. If the authors are going to use the results from their APOE2/2 vs. all genotypes comparison as a major justification of their conclusions, it is important to see the same comparisons for each other genotype (and not just each genotype vs. APOE3/3).
- 3) The results in Supplementary Fig. 10 showing that lipidated apoE2 does not bind to tau to a greater extent than other lipidated apoE isoforms is a major finding that greatly calls into question the authors' primary mechanistic speculation. The authors have done a satisfactory job of tamping down their mechanistic conclusions (although the word "likely" is still utilized on lines 77 and 258 to describe this mechanistic speculation). However, the authors still cite the increased interaction between non-lipidated apoE2 and tau as a plausible explanation for the increased tau pathology that they observed. Given that most apoE in the brain is lipidated, it is incumbent on the authors to either show a novel scenario where endogenously produced non-lipidated apoE interacts with tau or else remove this as a potential explanation. At the very least, the authors should move the results in Supplementary Fig. 10 into Fig. 3, so that the reader can clearly observe that lipidated apoE does not show this effect. Also, as a suggestion, the authors may want to consider the possibility that the increased levels of apoE protein observed in the APOE2 mice may itself be sufficient explanation for an increased interaction between apoE and tau in these mice (versus the original hypothesis that the disulfide bonds in apoE2 is mediating this effect).

4) In both the main text (line 112) and in Supplementary Fig. 7e, the authors describe "Alf1" mRNA levels. I believe the authors intended to say "Iba1" here and not "Alf1"

5) In lines 235 and 236, the authors misspell the word "neuroinflammation." An additional "n" should be added.

Reviewer #2 (Remarks to the Author):

The authors have adequately addressed this reviewer's concerns. The revised manuscript is much improved.

Reviewer #3 (Remarks to the Author):

Zhao and colleagues have generally been responsive to the reviewers concerns. Most of the major issues have been appropriately addressed and the manuscript is significantly improved. In particular, the authors have now included results demonstrating that the differences observed in the severity of tau pathology after AAV injection is not associated with different transduction levels between APOE2, APOE3 and APOE4-TR mice, and that the increased tau pathology detected in APOE2-TR mice is not due to a difference in neuronal survival as compared to APOE3-TR and APOE4-TR. Additionally, the revised paper now carefully states that, to some extent, the effects observed in apoE2-TR mice also tend to be observed in apoE3-TR mice.

Despite substantial improvement of the paper, one last concern still persists and need to be resolved. Indeed, the authors hypothesize that the impact of apoE on tau pathological changes may be due to the direct interaction between apoE and tau. Because no proper co-IP experiment has been performed, this conclusion is not completely demonstrated. The high molecular weight species observed both for tau and apoE can essentially correspond to self-association between apoE or tau molecules between themselves, without necessarily interacting with one another. In the case of apoE, the bands A and B could also correspond to the association of 3 and 4 apoE molecules. The use of the ECL system to develop the western blot membranes prevents a clear overlay of the band patterns between tau and apoE, and therefore lower the strength of the results. More importantly, in the case of the supplementary figure 10, it appears that the high molecular weight species A and B do not seem to be clearly visible in the blot for apoE. Overall, the assumption that apoE and tau directly interact is not proven and the data may be removed from the manuscript (which would not lower the impact of the other findings of the paper).

A few minor points remain to address:

- Line 94: "Consistently" should replace "Cosistently".
- P.6: The authors should notify that mild changes in behavior are also observed in the TauP310L-apoE3 mice as well in their conclusion sentence.
- Line 145: "extracting" should replace "extractinbg".

To summarize, the study proposed by Zhao and colleagues is of high relevance to the field and the findings very novel. The manuscript will be suitable for publication after resolving the last issues mentioned above.

Reviewer #4 (Remarks to the Author):

I have thoroughly re-read the paper as revised and I have also read completely the responses from the authors.

In my opinion, the authors have dealt well with all the criticisms, and at this point, I would have nothing to add.

This is a very important and provocative paper. I am sure that it will spawn some earthquakes! APOE2 is considered to be the golden goose and CW holds that if you have APOE2 you are blessed indeed and you will live an extended life while your brain is protected from many things. This paper flies in the face of 25 yrs of such claims. The paper is extremely important since many groups are trying to deliver APOE2 with the notion that it will prevent AD. Not so likely now that we see the Bu data.

Re: NCOMMS-17-32638

Thank you for the re-review of our manuscript entitled “*APOE* ϵ 2 is associated with increased risk for primary tauopathy”. We are glad that the reviewers remain enthusiastic about our work. We also appreciate the specific comments and suggestions from the Reviewers #1 and #3 to further improve our manuscript. To address these, we have now revised the Figure 3 and Supplementary Figure 10 and clarifying other concerns.

Below are our specific point-by-point responses to Reviewers’ comments/concerns:

Reviewer #1: In the revised manuscript, the authors have gone to great lengths to address reviewers concerns and have provided detailed and clear explanations for all of the changes that were made to the original manuscript. While these efforts are appreciated and the manuscript is improved, there remain a few points of concern. Most importantly, while the data and conclusions are interesting, the major finding that APOE2 mice display more tau pathology using this AAV-P301L injection model is not as impactful as the title of the paper implies. Most importantly, this model is unique in that the virus-induced P301L does not cause neurodegeneration, which is a major feature of most tauopathy mouse models and of course, of human tauopathies. Therefore, while the ability of APOE2 to increase tau aggregation and phosphorylation is intriguing, these studies may be unintentionally missing a more important contribution to tauopathy played by APOE (and most notably APOE4) in terms of their effects on neurodegeneration, as were highlighted by the recent Shi et al. paper from the Holtzman lab. Furthermore, while there are interesting points of corroboration within the human data, I have concerns that the paper is making a bolder statement about APOE2’s contribution to human tauopathies than the data actually reveals. This is certainly interesting work that raises very important questions about the role of APOE2 in tau pathology, but the actual data may not merit the level of attention that this story is likely going to garner if it is published in Nature Communications.

---- We thank the Reviewer’s comments. Please find our point-by-point responses as follows.

I recommend the following revisions:

- 1) The title should be modified to highlight the role of APOE2 in increasing tau pathology, as opposed to increasing the “risk for primary tauopathy.” As is, the title makes it sound like APOE2 has been discovered as a risk factor gene for tauopathy, which is clearly not the case. The authors did find an association between APOE2/2 carriers and PSP, and an increase in tau pathology in APOE2/3 carriers with PSP. However, the fact that APOE2/3 carriers also appear to

have a lower risk of developing PSP (despite having greater tau burden than APOE3/3 carriers with PSP) is somewhat contradictory. I also question whether the number of APOE2/2 carriers used in this study is sufficient to draw definitive conclusions about the susceptibility of APOE2/2 carriers to tauopathies.

---- We thank the Reviewer's suggestion. We agree with the Reviewer that the total number of APOE2/2 carriers is small in our PSP cohort (13/994 cases) due to the low frequency of APOE2/2 in the general population compared with other APOE genotypes. Also, it is not clear why APOE2/3 genotype trends to lower the PSP risk but increasing tau burden compared with APOE3/3 genotype. To draw a definitive conclusion on whether APOE2/2 is a risk gene for PSP, and whether APOE2/3 has the opposite effect on PSP as APOE2/2 genotype, we and others do need to validate these results in different cohorts in future studies. Nonetheless, we agree with the reviewer that the primary finding of this manuscript is the increased tau pathology by APOE2 both in human postmortem brains and in mouse models. As such, we agree with the reviewer to modify our title to emphasize the effect of APOE2 on tau pathology rather than risk for PSP. Our revised title is "APOE ϵ 2 is associated with increased tau pathology in primary tauopathy".

2) In Table 2, the authors should include the data comparing each genotype to all other genotypes. If the authors are going to use the results from their APOE2/2 vs. all genotypes comparison as a major justification of their conclusions, it is important to see the same comparisons for each other genotype (and not just each genotype vs. APOE3/3).

---- We thank the Reviewer for the opportunity to address this issue. In our original manuscript we had compared each genotype vs. all other genotypes in Table 2; however, in order to address a comment by a different Reviewer (Reviewer #3), we modified our analyses to make comparisons of each genotype vs. the theoretical neutral E3/E3 genotype which results in more unbiased association estimates. However, this approach also results in a reduction in sample size (as only the given genotype and E3/E3 are being examined for a given comparison). Therefore, if there is only one strong APOE risk/protective genotype, this approach can unnecessarily reduce the power for examining this specific risk genotype. Based on our data, there does in fact appear to be only one strong APOE risk/protective genotype – the E2/E2 genotype which increases the odds of PSP by more than 4-fold (Odds ratio=4.38). All other association odds ratios regarding association with PSP are fairly modest, ranging between 0.67 and 1.32. Therefore, to maximize power to detect an association with this one strong PSP risk genotype, we combined all other non-E2/E2 genotypes for comparison with the E2/E2 genotype, and observed a nearly identical odds ratio for association with PSP (4.41) but a stronger p-value (P=0.0057). In our opinion, given the strong association between the E2/E2 genotype and risk of PSP, it would not be reasonable to combine this genotype with other genotypes. For these reasons, we prefer to only analyze E2/E2 vs all other genotypes for association with risk of PSP (or CBD).

3) The results in Supplementary Fig. 10 showing that lipidated apoE2 does not bind to tau to a greater extent than other lipidated apoE isoforms is a major finding that greatly calls into question the authors' primary mechanistic speculation. The authors have done a satisfactory job of tamping down their mechanistic conclusions (although the word "likely" is still utilized on lines 77 and 258 to describe this mechanistic speculation). However, the authors still cite the increased interaction between non-lipidated apoE2 and tau as a plausible explanation for the increased tau pathology that they observed. Given that most apoE in the brain is lipidated, it is

incumbent on the authors to either show a novel scenario where endogenously produced non-lipidated apoE interacts with tau or else remove this as a potential explanation. At the very least, the authors should move the results in Supplementary Fig. 10 into Fig. 3, so that the reader can clearly observe that lipidated apoE does not show this effect. Also, as a suggestion, the authors may want to consider the possibility that the increased levels of apoE protein observed in the APOE2 mice may itself be sufficient explanation for an increased interaction between apoE and tau in these mice (versus the original hypothesis that the disulfide bonds in apoE2 is mediating this effect).

---- We thank the Reviewer's suggestions. We agree with the Reviewer that the interaction of apoE and tau might not be the primary mechanism to explain the apoE isoform effect on tau pathology according to our current data. Taking the suggestion from both Reviewers (Reviewers #1 and #3), we have now moved the tau and apoE interaction data into Supplementary Fig 10, and discussed the effect of such interaction as a potential mechanism (*Page 11, Line 7-10*).

4) In both the main text (line 112) and in Supplementary Fig. 7e, the authors describe "Alf1" mRNA levels. I believe the authors intended to say "Iba1" here and not "Alf1"

---- We thank the Reviewer for the careful review. The Iba1 protein is encoded by the gene *Aif1* (allograft inflammatory factor 1). We apologize misspelling *Aif1* as *Alf1* in our manuscript. We have now corrected the spelling in both the main text and Supplementary Fig 7e.

5) In lines 235 and 236, the authors misspell the word "neuroinflammation." An additional "m" should be added.

---- We thank the Reviewer for the careful review and have now changed the spelling accordingly.

Reviewer #2: The authors have adequately addressed this reviewer's concerns. The revised manuscript is much improved.

---- We thank the Reviewer for the satisfactory comment on our previous revision efforts. As such, no further revision is needed from this reviewer.

Reviewer #3: Zhao and colleagues have generally been responsive to the reviewers concerns. Most of the major issues have been appropriately addressed and the manuscript is significantly improved. In particular, the authors have now included results demonstrating that the differences observed in the severity of tau pathology after AAV injection is not associated with different transduction levels between APOE2, APOE3 and APOE4-TR mice, and that the increased tau pathology detected in APOE2-TR mice is not due to a difference in neuronal survival as compared to APOE3-TR and APOE4-TR. Additionally, the revised paper now carefully states that, to some extent, the effects observed in apoE2-TR mice also tend to be observed in apoE3-TR mice. Despite substantial improvement of the paper, one last concern still persists and need to be resolved. Indeed, the authors hypothesize that the impact of apoE on tau pathological changes may be due to the direct interaction between apoE and tau. Because no proper co-IP experiment has been performed, this conclusion is not completely demonstrated. The high

molecular weight species observed both for tau and apoE can essentially correspond to self-association between apoE or tau molecules between themselves, without necessarily interacting with one another. In the case of apoE, the bands A and B could also correspond to the association of 3 and 4 apoE molecules. The use of the ECL system to develop the western blot membranes prevents a clear overlay of the band patterns between tau and apoE, and therefore lower the strength of the results. More importantly, in the case of the supplementary figure 10, it appears that the high molecular weight species A and B do not seem to be clearly visible in the blot for apoE. Overall, the assumption that apoE and tau directly interact is not proven and the data may be removed from the manuscript (which would not lower the impact of the other findings of the paper).

---- We thank the Reviewer’s positive comments and thoughtful suggestions. We used Odyssey infrared imaging system (LI-COR) to obtain images of the Western blots for tau and apoE binding experiments; however, both tau and apoE antibodies were from mouse such that we could not blot with the two antibodies at the same time to observe the overlaid bands. We did try the co-IP experiment for tau and apoE interaction with mouse brain lysates. Unfortunately, we did not detect tau protein by pulling down apoE with K74 apoE antibody in these brain lysates (please refer to the figure as follow). We thought that the interaction between tau and apoE might be too weak to be successfully detected by the co-IP. As such, we could not make a definitive conclusion on the interaction of tau and apoE with robust *in vivo* or *in vitro* experiments. Several papers had reported the differential binding property between tau and apoE3 or apoE4 (recombinant or lipidated); however, our manuscript is the first to compare the binding between tau and all the three apoE isoforms, including apoE2. Considering this value of the data, and also taking this Reviewer and the Reviewer #1’s suggestions together, we agree to move the tau and apoE interaction data into Supplementary Fig 10, and not emphasize this interaction as a conclusive mechanism.

A few minor points remain to address:

- Line 94: “Consistently” should replace “Cosistently”.

---- We thank the Reviewer for the careful review and have now changed the spelling accordingly.

- P.6: The authors should notify that mild changes in behavior are also observed in the TauP310L-apoE3 mice as well in their conclusion sentence.

---- We thank the Reviewer’s suggestion and have modified our conclusions accordingly (*Page 6, Line 9*).

- Line 145: “extracting” should replace “extractinbg”.

---- We thank the Reviewer for careful review and have changed the spelling accordingly.

To summarize, the study proposed by Zhao and colleagues is of high relevance to the field and the findings very novel. The manuscript will be suitable for publication after resolving the last issues mentioned above.

---- We thank the Reviewer for recognizing the significance and impact of our work.

Reviewer #4: I have thoroughly re-read the paper as revised and I have also read completely the responses from the authors. In my opinion, the authors have dealt well with all the criticisms, and at this point, I would have nothing to add. This is a very important and provocative paper. I am sure that it will spawn some earthquakes! APOE2 is considered to be the golden goose and CW holds that if you have APOE2 you are blessed indeed and you will live an extended life while your brain is protected from many things. This paper flies in the face of 25 yrs of such claims. The paper is extremely important since many groups are trying to deliver APOE2 with the notion that it will prevent AD. Not so likely now that we see the Bu data.

---- We greatly appreciate the Reviewer for the overall enthusiasm and satisfaction of our work!

Thank you for your valuable inputs and guidance during the revision process of this manuscript! We believe that we have addressed all of reviewers' suggestions and comments to further improve our work. We trust that you now find our further revised manuscript acceptable for publication in the *Nature Communications*.